# Geometry-Aware Gradient Algorithms for Neural Architecture Search

**Liam Li**[*1]**, Mikhail Khodak**[*2]**, Maria-Florina Balcan**[2]**, and Ameet Talwalkar**[1,2]

[1] Determined AI, [2] Carnegie Mellon University, [*] denotes equal contribution

`me@liamcli.com,khodak@cmu.edu,ninamf@cs.cmu.edu,talwalkar@cmu.edu`

## Abstract

Recent state-of-the-art methods for neural architecture search (NAS) exploit gradient-based optimization by relaxing the problem into continuous optimization over architectures and shared-weights, a noisy process that remains poorly understood. We argue for the study of single-level empirical risk minimization to understand NAS with weight-sharing, reducing the design of NAS methods to devising optimizers and regularizers that can quickly obtain high-quality solutions to this problem. Invoking the theory of mirror descent, we present a geometry-aware framework that exploits the underlying structure of this optimization to return sparse architectural parameters, leading to simple yet novel algorithms that enjoy fast convergence guarantees and achieve state-of-the-art accuracy on the latest NAS benchmarks in computer vision. Notably, we exceed the best published results for both CIFAR and ImageNet on both the DARTS search space and NAS-Bench-201; on the latter we achieve near-oracle-optimal performance on CIFAR-10 and CIFAR-100. Together, our theory and experiments demonstrate a principled way to co-design optimizers and continuous relaxations of discrete NAS search spaces.

## 1 Introduction

Neural architecture search has become an important tool for automating machine learning (ML) but can require hundreds of thousands of GPU-hours to train. Recently, *weight-sharing* approaches have achieved state-of-the-art performance while drastically reducing the computational cost of NAS to just that of training a single *shared-weights network* (Pham et al., 2018; Liu et al., 2019). Methods such as DARTS (Liu et al., 2019), GDAS (Dong & Yang, 2019), and many others (Pham et al., 2018; Zheng et al., 2019; Yang et al., 2020; Xie et al., 2019; Liu et al., 2018; Laube & Zell, 2019; Cai et al., 2019; Akimoto et al., 2019; Xu et al., 2020) combine weight-sharing with a continuous relaxation of the discrete search space to allow cheap gradient updates, enabling the use of popular optimizers. However, despite some empirical success, weight-sharing remains poorly understood and has received criticism due to (1) rank-disorder (Yu et al., 2020; Zela et al., 2020b; Zhang et al., 2020; Pourchot et al., 2020), where the shared-weights performance is a poor surrogate of standalone performance, and (2) poor results on recent benchmarks (Dong & Yang, 2020; Zela et al., 2020a).

Motivated by the challenge of developing simple and efficient methods that achieve state-of-the-art performance, we study how to best handle the goals and optimization objectives of NAS. We start by observing that weight-sharing subsumes architecture hyperparameters as another set of learned parameters of the shared-weights network, in effect extending the class of functions being learned. This suggests that a reasonable approach towards obtaining high-quality NAS solutions is to study how to regularize and optimize the empirical risk over this extended class. While many regularization approaches have been implicitly proposed in recent NAS efforts, we focus instead on the question of optimizing architecture parameters, which may not be amenable to standard procedures such as SGD that work well for standard neural network weights. In particular, to better-satisfy desirable properties such as generalization and sparsity of architectural decisions, we propose to constrain architecture parameters to the simplex and update them using exponentiated gradient, which has favorable convergence properties due to the underlying problem structure. Theoretically, we draw upon the mirror descent meta-algorithm (Nemirovski & Yudin, 1983; Beck & Teboulle, 2003) to give convergence guarantees when using any of a broad class of such *geometry-aware* gradient methods to optimize the weight-sharing objective; empirically, we show that our solution leads to strong improvements on several NAS benchmarks. We summarize these contributions below:

1. We argue for studying NAS with weight-sharing as a single-level objective over a structured function class in which architectural decisions are treated as learned parameters rather than hyper-parameters. Our setup clarifies recent concerns about rank disorder and makes clear that proper regularization and optimization of this objective is critical to obtaining high-quality solutions.

2. Focusing on optimization, we propose to improve existing NAS algorithms by re-parameterizing architecture parameters over the simplex and updating them using exponentiated gradient, a variant of mirror descent that converges quickly over this domain and enjoys favorable sparsity properties. This simple modification—which we call the **Geometry-Aware Exponentiated Algorithm (GAEA)**—is easily applicable to numerous methods, including first-order DARTS Liu et al. (2019), GDAS Dong & Yang (2019), and PC-DARTS (Xu et al., 2020).

3. To show correctness and efficiency of our scheme, we prove polynomial-time stationary-point convergence of block-stochastic mirror descent—a family of geometry-aware gradient algorithms that includes GAEA—over a continuous relaxation of the single-level NAS objective. To the best of our knowledge these are the first finite-time convergence guarantees for gradient-based NAS.

4. We demonstrate that GAEA improves upon state-of-the-art methods on three of the latest NAS benchmarks for computer vision. Specifically, we beat the current best results on NAS-Bench-201 (Dong & Yang, 2020) by 0.18% on CIFAR-10, 1.59% on CIFAR-100, and 0.82% on ImageNet-16-120; we also outperform the state-of-the-art on the DARTS search space Liu et al. (2019), for both CIFAR-10 and ImageNet, and match it on NAS-Bench-1Shot1 (Zela et al., 2020a).[1]

**Related Work**. Most optimization analyses of NAS show monotonic improvement (Akimoto et al., 2019), asymptotic guarantees (Yao et al., 2020), or bounds on auxiliary quantities disconnected from any objective (Noy et al., 2019; Nayman et al., 2019; Carlucci et al., 2019). In contrast, we prove polynomial-time stationary-point convergence on a single-level objective for weight-sharing NAS, so far only studied empirically (Xie et al., 2019; Li et al., 2019). Our results draw upon the mirror descent meta-algorithm (Nemirovski & Yudin, 1983; Beck & Teboulle, 2003) and extend recent nonconvex convergence results Zhang & He (2018) to handle alternating descent. While there exist related results (Dang & Lan, 2015) the associated guarantees do not hold for the algorithms we propose. Finally, we note that a variant of GAEA that modifies first-order DARTS is related to XNAS (Nayman et al., 2019), whose update also involves exponentiated gradient; however, GAEA is simpler and easier to implement.[2] Furthermore, the regret guarantees for XNAS do not relate to any meaningful performance measure for NAS such as speed or accuracy, whereas we guarantee convergence on the ERM objective.

## 2 THE WEIGHT-SHARING OPTIMIZATION PROBLEM

In supervised ML we have a dataset $T$ of labeled pairs $(x, y)$ drawn from a distribution $\mathcal{D}$ over input/output spaces $X$ and $Y$. The goal is to use $T$ to search a function class $H$ for $h_\mathbf{w} : X \mapsto Y$ parameterized by $\mathbf{w} \in \mathbb{R}^d$ that has low expected test loss $\ell(h_\mathbf{w}(x), y)$ when using $x$ to predict the associated $y$ on unseen samples drawn from $D$, as measured by some loss $\ell : Y \times Y \mapsto [0, \infty)$. A common way to do so is by approximate (regularized) empirical risk minimization (ERM), i.e. finding $\mathbf{w} \in \mathbb{R}^d$ with the smallest average loss over $T$, via some iterative method Alg, e.g. SGD.

### 2.1 THE BENEFITS AND CRITICISMS OF WEIGHT-SHARING FOR NAS

NAS is often viewed as hyperparameter optimization on top of Alg, with each architecture $a \in \mathcal{A}$ corresponding to a function class $H_a = \{h_{\mathbf{w},a} : X \mapsto Y, \mathbf{w} \in \mathbb{R}^d\}$ to be selected by using validation data $V \subset X \times Y$ to evaluate the predictor obtained by fixing $a$ and doing approximate ERM over $T$:

$$\min_{a \in \mathcal{A}} \sum_{(x,y) \in V} \ell(h_{\mathbf{w}_a,a}(x), y) \qquad \text{s.t.} \qquad \mathbf{w}_a = \text{Alg}(T, a) \tag{1}$$

Since training individual sets of weights for any sizeable number of architectures is prohibitive, weight-sharing methods instead use a single set of shared weights to obtain validation signal about many architectures at once. In its most simple form, RS-WS (Li & Talwalkar, 2019), these weights

---

[1]Code to obtain these results has been made available in the supplementary material.

[2]XNAS code does not implement search and, as with previous efforts (Li et al., 2019, OpenReview), we cannot reproduce results after correspondence with the authors. XNAS's best architecture achieves an average test error of 2.70% under the DARTS evaluation, while GAEA achieves 2.50%. For details see Appendix C.4.

are trained to minimize a *non-adaptive* objective, $\min_{\mathbf{w} \in \mathbb{R}^d} \mathbb{E}_a \sum_{(x,y) \in T} \ell(h_{\mathbf{w}_a,a}(x), y)$, where the expectation is over a fixed distribution over architectures $\mathcal{A}$. The final architecture $a$ is then chosen to maximize the outer (validation) objective in (1) subject to $\mathbf{w}_a = \mathbf{w}$. More frequently used is a *bilevel* objective over some continuous relaxation $\Theta$ of the architecture space $\mathcal{A}$, after which a valid architecture is obtained via a discretization step $\mathrm{Map} : \Theta \mapsto \mathcal{A}$ (Pham et al., 2018; Liu et al., 2019):

$$\min_{\theta \in \Theta} \sum_{(x,y) \in V} \ell(h_{\mathbf{w},\theta}(x), y) \qquad \text{s.t.} \qquad \mathbf{w} \in \arg\min_{\mathbf{u} \in \mathbb{R}^d} \sum_{(x,y) \in T} \ell(h_{\mathbf{u},\theta}(x), y) \tag{2}$$

This objective is not significantly different from (2), since $\mathrm{Alg}(T, a)$ approximately minimizes the empirical risk w.r.t. $T$; the difference is replacing discrete architectures with relaxed *architecture parameters* $\theta \in \Theta$, w.r.t. which we can take derivatives of the outer objective. This allows (2) to be approximated via alternating gradient updates w.r.t. $\mathbf{w}$ and $\theta$. Relaxations can be *stochastic*, so that $\mathrm{Map}(\theta)$ is a sample from a $\theta$-parameterized distribution (Pham et al., 2018; Dong & Yang, 2019), or a *mixture*, in which case $\mathrm{Map}(\theta)$ selects architectural decisions with the highest weight in a convex combination given by $\theta$ (Liu et al., 2019). We overview this in more detail in Appendix A.

While weight-sharing significantly shortens search (Pham et al., 2018), it draws two main criticisms:

- Rank disorder: this describes when the rank of an architecture $a$ according to the validation risk evaluated with fixed shared weights $\mathbf{w}$ is poorly correlated with the one using "standalone" weights $\mathbf{w}_a = \mathrm{Alg}(T, a)$. This causes suboptimal architectures to be selected after shared weights search (Yu et al., 2020; Zela et al., 2020b; Zhang et al., 2020; Pourchot et al., 2020).
- Poor performance: weight-sharing can converge to degenerate architectures (Zela et al., 2020a) and is outperformed by regular hyperparameter tuning on NAS-Bench-201 (Dong & Yang, 2020).

## 2.2 SINGLE-LEVEL NAS AS A BASELINE OBJECT OF STUDY

Why are we able to apply weight-sharing to NAS? The key is that, unlike regular hyperparameters such as step-size, *architectural* hyperparameters directly affect the loss function without requiring a dependent change in the model weights $\mathbf{w}$. Thus we can distinguish architectures without retraining simply by changing architectural decisions. Besides enabling weight-sharing, this point reveals that the goal of NAS is perhaps better viewed as a regular learning problem over an extended class $H_{\mathcal{A}} = \bigcup_{a \in \mathcal{A}} H_a = \{h_{\mathbf{w},a} : X \mapsto Y, \mathbf{w} \in \mathbb{R}^d, a \in \mathcal{A}\}$ that subsumes the architectural decisions as parameters of a larger model class, an unrelaxed "supernet." The natural approach to solving this is by approximate empirical risk minimization, e.g. by approximating continuous objective below on the right using a gradient algorithm and passing the output $\theta$ through $\mathrm{Map}$ to obtain a valid architecture:

$$\underbrace{\min_{\mathbf{w} \in \mathbb{R}^d, a \in \mathcal{A}} \sum_{(x,y) \in T} \ell(h_{\mathbf{w},a}(x), y)}_{\text{discrete (unrelaxed) supernet (NAS ERM)}} \qquad \underbrace{\min_{\mathbf{w} \in \mathbb{R}^d, \theta \in \Theta} \sum_{(x,y) \in T} \ell(h_{\mathbf{w},\theta}(x), y)}_{\text{continuous relaxation (supernet ERM)}} \tag{3}$$

Several works have optimized this single-level objective as an alternative to bilevel (2) (Xie et al., 2019; Li et al., 2019). We argue for its use as the baseline object of study in NAS for three reasons:

1. As discussed above, it is the natural first approach to solving the statistical objective of NAS: finding a good predictor $h_{\mathbf{w},a} \in H_{\mathcal{A}}$ in the extended function class over architectures and weights.
2. The common alternating gradient approach to the bilevel problem (2) is in practice very similar to alternating block approaches to ERM (3); as we will see, there are established ways of analyzing such methods for the latter objective, while for the former convergence is known only under very strong assumptions such as uniqueness of the inner minimum (Franceschi et al., 2018).
3. While less frequently used in practice than bilevel, single-level optimization can be very effective: we use it to achieve new state-of-the-art results on NAS-Bench-201 (Dong & Yang, 2020).

Understanding NAS as single-level optimization—the usual deep learning setting—makes weight-sharing a natural, not surprising, approach. Furthermore, for methods—both single-level and bilevel—that adapt architecture parameters during search, it suggests that we need not worry about rank disorder as long as we can use optimization to find a single feasible point that generalizes well; we explicitly do *not* need a ranking. Non-adaptive methods such as RS-WS still do require rank correlation to select good architectures after search, but they are explicitly *not* changing $\theta$ and so have no variant solving (3). The single-level formulation thus reduces search method design to well-studied questions of how to best regularize and optimize ERM. While there are many techniques for regularizing weight-sharing—including partial channels (Xu et al., 2020) and validation Hessian penalization (Zela et al., 2020a)—we focus on the second question of optimization.

## 3 GEOMETRY-AWARE GRADIENT ALGORITHMS

We seek to minimize the (possibly regularized) empirical risk $f(\mathbf{w}, \theta) = \frac{1}{|T|} \sum_{(x,y) \in T} \ell(h_{\mathbf{w},\theta}(x), y)$ over shared-weights $\mathbf{w} \in \mathbb{R}^d$ and architecture parameters $\theta \in \Theta$. Assuming we have noisy gradients of $f$ w.r.t. $\mathbf{w}$ or $\theta$ at any point $(\mathbf{w}, \theta) \in \mathbb{R}^d \times \Theta$—i.e. $\tilde{\nabla}_{\mathbf{w}} f(\mathbf{w}, \theta)$ or $\tilde{\nabla}_\theta f(\mathbf{w}, \theta)$ satisfying $\mathbb{E}\tilde{\nabla}_{\mathbf{w}} f(\mathbf{w}, \theta) = \nabla_{\mathbf{w}} f(\mathbf{w}, \theta)$ or $\mathbb{E}\tilde{\nabla}_\theta f(\mathbf{w}, \theta) = \nabla_\theta f(\mathbf{w}, \theta)$, respectively—our goal is a point where $f$, or at least its gradient, is small, while taking as few gradients as possible. Our main complication is that architecture parameters lie in a constrained, non-Euclidean domain $\Theta$. Most search spaces $\mathcal{A}$ are product sets of categorical decisions—which operation $o \in O$ to use at edge $e \in E$—so the natural relaxation is a product of $|E|$ $|O|$-simplices. However, NAS methods often re-parameterize $\Theta$ to be unconstrained using a softmax and then SGD or Adam (Kingma & Ba, 2015). Is there a better parameterization-algorithm co-design? We consider a *geometry-aware* approach that uses mirror descent to design NAS methods with better properties depending on the domain; a key desirable property is to return *sparse* architectural parameters to reduce loss from post-search discretization.

### 3.1 BACKGROUND ON MIRROR DESCENT

Mirror descent has many formulations (Nemirovski & Yudin, 1983; Beck & Teboulle, 2003; Shalev-Shwartz, 2011); the *proximal* starts by noting that, in the unconstrained case, an SGD update at a point $\theta \in \Theta = \mathbb{R}^k$ given gradient estimate $\tilde{\nabla} f(\theta)$ with step-size $\eta > 0$ is equivalent to

$$\theta - \eta \tilde{\nabla} f(\theta) \quad = \quad \operatorname*{arg\,min}_{\mathbf{u} \in \mathbb{R}^k} \quad \eta \tilde{\nabla} f(\theta) \cdot \mathbf{u} \quad + \quad \frac{1}{2}\|\mathbf{u} - \theta\|_2^2 \tag{4}$$

Here the first term aligns the output with the gradient while the second (proximal) term regularizes for closeness to the previous point as measured by the Euclidean distance. While the SGD update has been found to work well for unconstrained high-dimensional optimization, e.g. deep nets, this choice of proximal regularization may be sub-optimal over a constrained space with sparse solutions. The canonical such setting is optimization over the unit simplex, i.e. when $\Theta = \{\theta \in [0,1]^k : \|\theta\|_1 = 1\}$. Replacing the $\ell_2$-regularizer in Equation 4 by the relative entropy $\mathbf{u} \cdot (\log \mathbf{u} - \log \theta)$, i.e. the KL-divergence, yields the exponentiated gradient (EG) update ($\odot$ denotes element-wise product):

$$\theta \odot \exp(-\eta \tilde{\nabla} f(\theta)) \quad \propto \quad \operatorname*{arg\,min}_{\mathbf{u} \in \Theta} \quad \eta \tilde{\nabla} f(\theta) \cdot \mathbf{u} \quad + \quad \mathbf{u} \cdot (\log \mathbf{u} - \log \theta) \tag{5}$$

Note that the full EG update is obtained by $\ell_1$-normalizing the l.h.s. It is well-known that EG over the $k$-dimensional simplex requires only $\mathcal{O}(\log k)/\varepsilon^2$ iterations to achieve a function value $\varepsilon$-away from optimal (Beck & Teboulle, 2003, Theorem 5.1), compared to the $O(k/\varepsilon^2)$ guarantee of gradient descent. This nearly dimension-independent iteration complexity is achieved by choosing a regularizer—the KL divergence—well-suited to the underlying geometry—the simplex. More generally, mirror descent is specified by a distance-generating function (DGF) $\phi$ that is strongly-convex w.r.t. some norm. $\phi$ induces a *Bregman divergence* $D_\phi(\mathbf{u}\|\mathbf{v}) = \phi(\mathbf{u}) - \phi(\mathbf{v}) - \nabla\phi(\mathbf{v}) \cdot (\mathbf{u} - \mathbf{v})$ (Bregman, 1967), a notion of distance on $\Theta$ that acts as a regularizer in the mirror descent update:

$$\operatorname*{arg\,min}_{\mathbf{u} \in \Theta} \quad \eta \tilde{\nabla} f(\theta) \cdot \mathbf{u} \quad + \quad D_\phi(\mathbf{u}\|\theta) \tag{6}$$

For example, to recover SGD (4) we set $\phi(\mathbf{u}) = \frac{1}{2}\|\mathbf{u}\|_2^2$, which is strongly-convex w.r.t. the Euclidean norm, while EG (5) is recovered by setting $\phi(\mathbf{u}) = \mathbf{u} \cdot \log \mathbf{u}$, strongly-convex w.r.t. the $\ell_1$-norm.

### 3.2 BLOCK-STOCHASTIC MIRROR DESCENT

In the previous section we saw how mirror descent can perform better over certain geometries such as the simplex. However, in weight-sharing we are interested in optimizing over a hybrid geometry containing both the shared weights in an unconstrained Euclidean space and the architecture parameters in a non-Euclidean domain. Thus we focus on optimization over two blocks: shared weights $\mathbf{w} \in \mathbb{R}^d$ and architecture parameters $\theta \in \Theta$, the latter associated with a DGF $\phi$ that is strongly-convex w.r.t. some norm $\|\cdot\|$. In NAS a common approach is to perform alternating gradient steps on each domain; for example, both ENAS (Pham et al., 2018) and first-order DARTS (Liu et al., 2019) alternate between SGD on the shared weights and Adam on architecture parameters. This approach is encapsulated in the block-stochastic algorithm described in Algorithm 1, which at each step chooses one block at random to update using mirror descent (recall that SGD is a variant) and after $T$ steps returns a random iterate. Algorithm 1 generalizes the single-level variant of both ENAS and first-order DARTS if SGD is used to update $\theta$ instead of Adam, with some mild caveats: in practice blocks are picked cyclically and the algorithm returns the last iterate, not a a random one. To analyze the convergence of Algorithm 1 we first state some regularity assumptions on the function:

---

**Algorithm 1:** Block-stochastic mirror descent optimization of a function $f : \mathbb{R}^d \times \Theta \mapsto \mathbb{R}$.

---

**Input:** initialization $(\mathbf{w}^{(1)}, \theta^{(1)}) \in \mathbb{R}^d \times \Theta$, strongly-convex DGF $\phi : \Theta \mapsto \mathbb{R}$, number of iterations $T \geq 1$, step-size $\eta > 0$

**for** iteration $t = 1, \ldots, T$ **do**

    sample $b_t \sim \text{Unif}\{\mathbf{w}, \theta\}$                                          // randomly select update block

    **if** block $b_t = \mathbf{w}$ **then**

        $\mathbf{w}^{(t+1)} \leftarrow \mathbf{w}^{(t)} - \eta \tilde{\nabla}_{\mathbf{w}} f(\mathbf{w}^{(t)}, \theta^{(t)})$             // SGD update to shared weights

        $\theta^{(t+1)} \leftarrow \theta^{(t)}$                                 // no update to architecture params

    **else**

        $\mathbf{w}^{(t+1)} \leftarrow \mathbf{w}^{(t)}$                             // no update to shared weights

        $\theta^{(t+1)} \leftarrow \arg\min_{\mathbf{u} \in \Theta} \;\; \eta \tilde{\nabla}_\theta f(\mathbf{w}^{(t)}, \theta^{(t)}) \cdot \mathbf{u} \; + \; D_\phi(\mathbf{u} || \theta^{(t)})$     // update architecture params

**Output:** $(\mathbf{w}^{(r)}, \theta^{(r)})$ for $r \sim \text{Unif}\{1, \ldots, T\}$                     // return random iterate

---

**Assumption 1.** *Suppose $\phi$ is strongly-convex w.r.t. some norm $\| \cdot \|$ on a convex set $\Theta$ and the objective function $f : \mathbb{R}^d \times \Theta \mapsto [0, \infty)$ satisfies the following:*

*1.* **$\gamma$-relatively-weak-convexity:** *$f(\mathbf{w}, \theta) + \gamma \phi(\theta)$ is convex on $\mathbb{R}^d \times \Theta$ for some $\gamma > 0$.*

*2.* **gradient bound:** *$\mathbb{E}\|\tilde{\nabla}_{\mathbf{w}} f(\mathbf{w}, \theta)\|_2^2 \leq G_{\mathbf{w}}^2$ and $\mathbb{E}\|\tilde{\nabla}_\theta f(\mathbf{w}, \theta)\|_*^2 \leq G_\theta^2$ for some $G_{\mathbf{w}}, G_\theta \geq 0$.*

The second assumption is a standard bound on the gradient norm while the first is a generalization of smoothness that allows all smooth and some non-smooth non-convex functions (Zhang & He, 2018).

Our aim will be to show (first-order) $\varepsilon$-stationary-point convergence of Algorithm 1, a standard metric indicating that it has reached a point with no feasible descent direction, up to error $\varepsilon$; for example, in the unconstrained Euclidean case an $\varepsilon$-stationary-point is simply one where the gradient has squared-norm $\leq \varepsilon$. The number of steps required to obtain such a point thus measures how fast a first-order method terminates. Stationarity is also significant as a necessary condition for optimality.

In our case $\Theta$ may be constrained and so the gradient may never be small, thus necessitating a measure other than gradient norm. We use *Bregman stationarity* (Zhang & He, 2018, Equation 2.11), which measures stationary at a point $(\mathbf{w}, \theta)$ using the Bregman divergence between the point and its proximal map $\text{prox}_\lambda(\mathbf{w}, \theta) = \arg\min_{\mathbf{u} \in \mathbb{R}^d \times \Theta} \lambda f(\mathbf{u}) + D_{\ell_2, \phi}(\mathbf{u} || \mathbf{w}, \theta)$ for some $\lambda > 0$:

$$\Delta_\lambda(\mathbf{w}, \theta) = \frac{D_{\ell_2, \phi}(\mathbf{w}, \theta || \text{prox}_\lambda(\mathbf{w}, \theta)) + D_{\ell_2, \phi}(\text{prox}_\lambda(\mathbf{w}, \theta) || \mathbf{w}, \theta)}{\lambda^2} \tag{7}$$

Here $\lambda = \frac{1}{2\gamma}$ and the Bregman divergence $D_{\ell_2, \phi}$ is that of the DGF $\frac{1}{2}\|\mathbf{w}\|_2^2 + \phi(\theta)$ that encodes the geometry of the joint optimization domain over $\mathbf{w} \in \mathbb{R}^d$ and $\theta$; note that the dependence of the stationarity measure on $\gamma$ is standard (Dang & Lan, 2015; Zhang & He, 2018).

To understand why reaching a point $(\mathbf{w}, \theta)$ with small Bregman stationarity is a reasonable goal, note that the proximal operator $\text{prox}_\lambda$ has the property that its fixed points, i.e. those satisfying $(\mathbf{w}, \theta) = \text{prox}_\lambda(\mathbf{w}, \theta)$, correspond to points where $f$ has no feasible descent direction. Thus measuring how close $(\mathbf{w}, \theta)$ is to being a fixed point of $\text{prox}_\lambda$—as is done using the Bregman divergence in (7)—is a good measure of how far away the point is from being a stationary point of $f$. Finally, note that if $f$ is smooth, $\phi$ is Euclidean, and $\Theta$ is unconstrained—i.e. if we are running SGD over architecture parameters as well—then $\Delta_{\frac{1}{2\gamma}} \leq \varepsilon$ implies an $O(\varepsilon)$-bound on the squared gradient norm, recovering the standard definition of $\varepsilon$-stationarity. More intuition on proximal operators can be found in Parikh & Boyd (2013, Section 1.2), while further details on Bregman stationarity and how it relates to other notions of convergence can be found in Zhang & He (2018, Section 2.3).

The following result shows that Algorithm 1 needs polynomially many iterations to finds a point $(\mathbf{w}, \theta)$ with $\varepsilon$-small Bregman stationarity in-expectation:

**Theorem 1.** *Let $F = f(\mathbf{w}^{(1)}, \theta^{(1)})$ be the value of $f$ at initialization. Under Assumption 1, if we run Algorithm 1 for $T = \frac{16\gamma F}{\varepsilon^2}(G_{\mathbf{w}}^2 + G_\theta^2)$ iterations with step-size $\eta = \sqrt{\frac{4F}{\gamma(G_{\mathbf{w}}^2 + G_\theta^2)T}}$ then $\mathbb{E}\Delta_{\frac{1}{2\gamma}}(\mathbf{w}^{(r)}, \theta^{(r)}) \leq \varepsilon$. Here the expectation is over the randomness of the algorithm and gradients.*

The proof in the appendix follows from single-block analysis (Zhang & He, 2018, Theorem 3.1) and in fact holds for the general case of any number of blocks associated to any set of strongly-convex DGFs. Although there are prior results for the multi-block case (Dang & Lan, 2015), they do not hold for nonsmooth Bregman divergences such as the KL divergence needed for exponentiated gradient.

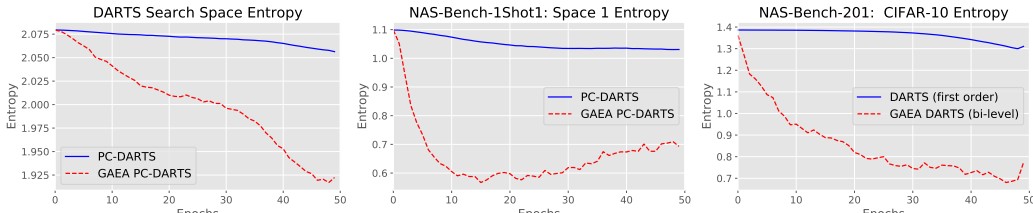

Figure 1: **Sparsity:** Evolution over search phase epochs of the average entropy of the operation-weights for GAEA and approaches it modifies when run on the DARTS search space (left), NAS-Bench-1Shot1 Search Space 1 (middle), and NASBench-201 on CIFAR-10 (right). GAEA reduces entropy much more quickly, allowing it to quickly obtain sparse architecture weights. This leads to both faster convergence to a single architecture and a lower loss when pruning at the end of search.

Thus Algorithm 1 returns an $\varepsilon$-stationary-point given $T = \mathcal{O}(G_{\mathbf{w}}^2 + G_\theta^2)/\varepsilon^2$ iterations, where $G_{\mathbf{w}}^2$ bounds the squared $\ell_2$-norm of the shared-weights gradient $\tilde{\nabla}_{\mathbf{w}}$ and $G_\theta^2$ bounds the squared magnitude of the architecture gradient $\tilde{\nabla}_\theta$, as measured by the *dual norm* $\| \cdot \|_*$ of $\| \cdot \|$. Only the last term $G_\theta$ is affected by our choice of DGF $\phi$. The DGF of SGD is strongly-convex w.r.t. the $\ell_2$-norm, which is its own dual, so $G_{\mathbf{w}}^2$ is defined via $\ell_2$. However, for EG the DGF $\phi(\mathbf{u}) = \mathbf{u} \cdot \log \mathbf{u}$ is strongly-convex w.r.t. the $\ell_1$-norm, whose dual is $\ell_\infty$. Since the $\ell_2$-norm of a $k$-dimensional vector can be $\sqrt{k}$ times its $\ell_\infty$-norm, picking this DGF can lead to better bound on $G_\theta$ and thus on the number of iterations.

### 3.3 GAEA: A GEOMETRY-AWARE EXPONENTIATED ALGORITHM

Equipped with these single-level guarantees, we turn to designing methods that can in-principle be applied to both the single-level and bilevel objectives, seeking parameterizations and algorithms that converge quickly and encourage favorable properties; in particular, we focus on returning architecture parameters that are *sparse* to reduce loss due to post-search discretization. EG is often considered to converge quickly to sparse solutions over the simplex (Bradley & Bagnell, 2008; Bubeck, 2019), which makes it a natural choice for the architecture update. We thus propose **GAEA**, a **Geometry-Aware Exponentiated Algorithm** in which operation weights on each edge are constrained to the simplex and trained using EG; as in DARTS, the shared weights $\mathbf{w}$ are trained using SGD. GAEA can be used as a simple, principled modification to the many NAS methods that treat architecture parameters $\theta \in \Theta = \mathbb{R}^{|E| \times |O|}$ as real-valued "logits" to be passed through a softmax to obtain mixture weights or probabilities for simplices over the operations $O$. Such methods include DARTS, PC-DARTS (Xu et al., 2020), and GDAS (Dong & Yang, 2019). To apply GAEA, first re-parameterize $\Theta$ to be the product set of $|E|$ simplices, each associated to an edge $(i, j) \in E$; thus $\theta_{i,j,o}$ corresponds directly to the weight or probability of operation $o \in O$ for edge $(i, j)$, not a logit. Then, given a stochastic gradient $\tilde{\nabla}_\theta f(\mathbf{w}^{(t)}, \theta^{(t)})$ and step-size $\eta > 0$, replace the architecture update by EG:

$$\tilde{\theta}^{(t+1)} \leftarrow \theta^{(t)} \odot \exp\left(-\eta\tilde{\nabla}_\theta f(\mathbf{w}^{(t)}, \theta^{(t)})\right) \qquad \text{(multiplicative update)}$$

$$\theta_{i,j,o}^{(t+1)} \leftarrow \frac{\tilde{\theta}_{i,j,o}^{(t+1)}}{\sum_{o' \in O} \tilde{\theta}_{i,j,o'}^{(t+1)}} \quad \forall\, o \in O, \, \forall\, (i, j) \in E \qquad \text{(simplex projection)} \tag{8}$$

These two simple modifications, *re-parameterization* and *exponentiation*, suffice to obtain state-of-the-art results on several NAS benchmarks, as shown in Section 4. Note that to obtain a bilevel algorithm we simply replace the gradient w.r.t. $\theta$ of the training loss with that of the validation loss.

GAEA is equivalent to Algorithm 1 with $\phi(\theta) = \sum_{(i,j) \in E} \sum_{o \in O} \theta_{i,j,o} \log \theta_{i,j,o}$, which is strongly-convex w.r.t. $\| \cdot \|_1/\sqrt{|E|}$ over the product of $|E|$ $|O|$-simplices. The dual is $\sqrt{|E|}\| \cdot \|_\infty$, so if $G_{\mathbf{w}}$ bounds the shared-weights gradient and we have an entry-wise bound on the architecture gradient then GAEA reach $\varepsilon$-stationarity in $\mathcal{O}(G_{\mathbf{w}}^2 + |E|)/\varepsilon^2$ iterations. This can be up to a factor $|O|$ improvement over SGD, either over the simplex or the logit space. In addition, GAEA encourages sparsity in the architecture weights by using a multiplicative update over simplices and not an additive update over $\mathbb{R}^{|E| \times |O|}$. Obtaining sparse architecture parameters is critical for good performance, both for the mixture relaxation, where it alleviates the effect of discretization on the validation loss, and for the stochastic relaxation, where it reduces noise when sampling architectures.

Table 1: **DARTS:** Comparison with SOTA NAS methods on the DARTS search space, plus three results on different search spaces with a similar number of parameters reported at the top for comparison. All evaluations and reported performances of models found on the DARTS search space use similar training routines; this includes auxiliary towers and cutout but no other modifications, e.g. label smoothing (Müller et al., 2019), AutoAugment (Cubuk et al., 2019), Swish (Ramachandran et al., 2017), Squeeze & Excite (Hu et al., 2018), etc. The specific training procedure we use is that of PC-DARTS, which differs slightly from the DARTS routine by a small change to the drop-path probability; PDARTS tunes both this and batch-size. Our results are averaged over 10 random seeds. Search cost is hardware-dependent; we used Tesla V100 GPUs. For more details see Tables 4 & 5.

| Search Method (source) | CIFAR-10 Error | | Search Cost (GPU Days) | ImageNet Error | | Search Cost (GPU Days) | method |
|---|---|---|---|---|---|---|---|
| | Best | Average | | top-1 | top-5 | | |
| NASNet-A[*] (Zoph et al., 2018) | - | 2.65 | 2000 | 26.0 | 8.4 | 1800 | RL |
| AmoebaNet-B[*] (Real et al., 2019) | - | $2.55 \pm 0.05$ | 3150 | 24.3 | 7.6 | 3150 | evolution |
| ProxylessNAS[*] (Cai et al., 2019) | 2.08 | - | 4 | 24.9 | 7.5 | 8.3 | gradient (WS) |
| ENAS (Pham et al., 2018) | 2.89 | - | 0.5 | - | - | - | RL (WS) |
| RS-WS[†] (Li & Talwalkar, 2019) | 2.71 | $2.85 \pm 0.08$ | 0.7 | - | - | - | random (WS) |
| ASNG (Akimoto et al., 2019) | - | $2.83 \pm 0.14$ | 0.1 | - | - | - | gradient (WS) |
| SNAS (Xie et al., 2019) | - | $2.85 \pm 0.02$ | 1.5 | 27.3 | 9.2 | 1.5 | gradient (WS) |
| DARTS (1st)[†] (Liu et al., 2019) | - | $3.00 \pm 0.14$ | 0.4 | - | - | - | gradient (WS) |
| DARTS (2nd)[†] (Liu et al., 2019) | - | $2.76 \pm 0.09$ | 1 | 26.7 | 8.7 | 4.0 | gradient (WS) |
| PDARTS (Chen et al., 2019) | 2.50 | - | 0.3 | 24.4 | 7.4 | 0.3 | gradient (WS) |
| PC-DARTS[†] (Xu et al., 2020) | - | $2.57 \pm 0.07$ | 0.1 | 24.2 | 7.3 | 3.8 | gradient (WS) |
| **GAEA** PC-DARTS[†] (ours) | 2.39 | $2.50 \pm 0.06$ | 0.1 | 24.0 | 7.3 | 3.8 | gradient (WS) |
| PC-DARTS[†] (Xu et al., 2020) | (search on CIFAR-10, train on ImageNet) | | | 25.1 | 7.8 | 0.1 | gradient (WS) |
| **GAEA** PC-DARTS[†] (ours) | (search on CIFAR-10, train on ImageNet) | | | 24.3 | 7.3 | 0.1 | gradient (WS) |

[*] Search space/backbone differ from the DARTS setting; we show results for networks with a comparable number of parameters.
[†] For fair comparison to other work, we show the search cost for training the shared-weights network with a single initialization.

## 4 EMPIRICAL RESULTS USING GAEA

We evaluate GAEA on three different computer vision benchmarks: the large and heavily studied search space from DARTS (Liu et al., 2019) and two smaller oracle evaluation benchmarks, NAS-Bench-1Shot1 (Zela et al., 2020a), and NAS-Bench-201 (Dong & Yang, 2020). NAS-Bench-1Shot1 differs from the others by applying operations per node instead of per edge, while NAS-Bench-201 differs by not requiring edge-pruning. Since GAEA can modify a variety of methods, e.g. DARTS, PC-DARTS (Xu et al., 2020), and GDAS (Dong & Yang, 2019), on each benchmark we start by evaluating the GAEA variant of the current best method on that benchmark. We show that despite the diversity of search spaces, GAEA improves upon this state-of-the-art across all three. Note that we use the same step-size for GAEA variants of DARTS/PC-DARTS and do not require weight-decay on architecture parameters. We defer experimental details and hyperparameter settings to the appendix and release all code, hyperparameters, and random seeds for reproducibility.

### 4.1 CONVERGENCE AND SPARSITY OF GAEA

We first examine the impact of GAEA on convergence and sparsity. Figure 1 shows the entropy of the operation weights averaged across nodes for a GAEA-variant and its base method across the three benchmarks, demonstrating that it decreases much faster for GAEA-modified approaches. This validates our expectation that GAEA encourages sparse architecture parameters, which should alleviate the mismatch between the continuously relaxed architecture parameters and the discrete architecture returned. Indeed, we find that post-search discretization on the DARTS search space causes the validation accuracy of the PC-DARTS supernet to drop from 72.17% to 15.27%, while for GAEA PC-DARTS the drop is only 75.07% to 33.23%; note that this is shared-weights accuracy, obtained *without* retraining the final network. The numbers demonstrate that GAEA both (1) achieves better supernet optimization of the weight-sharing objective and (2) suffers less due to discretization.

### 4.2 GAEA ON THE DARTS SEARCH SPACE

Here we evaluate GAEA on the task of designing CNN cells for CIFAR-10 (Krizhevsky, 2009) and ImageNet (Russakovsky et al., 2015) by using it to modify PC-DARTS (Xu et al., 2020), the current state-of-the-art method. We follow the same three stage process used by both DARTS and RS-WS for search and evaluation. Table 1 displays results on both datasets and demonstrates that GAEA's parameterization and optimization scheme improves upon PC-DARTS. In fact, GAEA PC-DARTS

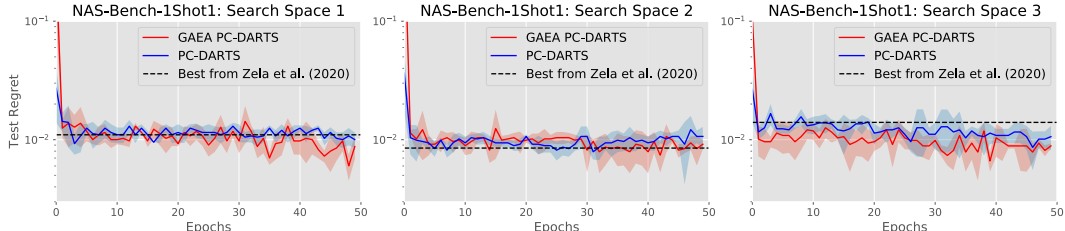

Figure 2: **NAS-Bench-1Shot1:** Online comparison of PC-DARTS and GAEA PC-DARTS in terms of the test regret at each epoch of shared-weights training, i.e. the difference between the ground truth test error of the proposed architecture and that of the best architecture in the search space. The dark lines indicate the mean of four random trials and the light colored bands ± one standard deviation. The dashed line is the final regret of the best weight-sharing method according to Zela et al. (2020b); note that in our reproduction PC-DARTS performed better than their evaluation on spaces 1 and 3.

outperforms all search methods except ProxylessNAS, which uses 1.5 times as many parameters on a different search space. Thus we improve the state-of-the-art on the DARTS search space. To meet a higher bar for reproducibility on CIFAR-10, in Appendix C we report "broad reproducibility" (Li & Talwalkar, 2019) by repeating our pipeline with new seeds. While GAEA PC-DARTS consistently finds good networks when selecting the best of four independent trials, multiple trials are required due to sensitivity to initialization, as is true for many approaches (Liu et al., 2019; Xu et al., 2020).

On ImageNet, we follow Xu et al. (2020) by using subsamples containing $10\%$ and $2.5\%$ of the training images from ILSVRC-2012 (Russakovsky et al., 2015) as training and validation sets, respectively. We fix architecture parameters for the first 35 epochs, then run GAEA PC-DARTS with step-size 0.1. All other hyperparameters match those of Xu et al. (2020). Table 1 shows the final performance of both the architecture found by GAEA PC-DARTS on CIFAR-10 and the one found directly on ImageNet when trained from scratch for 250 epochs using the same settings as Xu et al. (2020). GAEA PC-DARTS achieves a top-1 test error of $24.0\%$, which is state-of-the-art performance in the mobile setting when excluding additional training modifications, e.g. those in the caption. Additionally, the architecture found by GAEA PC-DARTS for CIFAR-10 and transferred achieves a test error of $24.2\%$, comparable to the $24.2\%$ error of the one found by PC-DARTS directly on ImageNet. Top architectures found by GAEA PC-DARTS are depicted in Figure 3 in the appendix.

### 4.3 GAEA ON NAS-BENCH-1SHOT1

NAS-Bench-1Shot1 (Zela et al., 2020a) is a subset of NAS-Bench-101 (Ying et al., 2019) that allows benchmarking weight-sharing methods on three search spaces over CIFAR-10 that differ in the number of nodes considered and the number of input edges per node. Of the weight-sharing methods benchmarked by Zela et al. (2020a), we found that PC-DARTS achieves the best performance on 2 of 3 search spaces, so we again evaluate GAEA PC-DARTS here. Figure 2 shows that GAEA PC-DARTS consistently finds better architectures on average than PC-DARTS and thus exceeds the performance of the best method from Zela et al. (2020a) on 2 of 3 search spaces. We hypothesize that the benefits of GAEA are limited here due to the near-saturation of NAS methods. In particular, existing methods obtain within $1\%$ test error of the top network in each space, while the latters' test errors when evaluated with different initializations are $0.37\%$, $0.23\%$ and $0.19\%$, respectively.

### 4.4 GAEA ON NAS-BENCH-201

NAS-Bench-201 has one search space on three datasets—CIFAR-10, CIFAR-100, and ImageNet-16-120—that includes 4-node architectures with an operation from $O = \{$none, skip connect, 1x1 convolution, 3x3 convolution, 3x3 avg pool$\}$ on each edge, yielding 15625 possible networks. Dong & Yang (2020) report results for several algorithms in the *transfer NAS* setting, where search is conducted on CIFAR-10 and the resulting networks are trained on a possibly different target dataset. Table 2 reports a subset of these results alongside evaluations of our implementation of several existing and GAEA-modified NAS methods in both the transfer and direct setting. Both the results from Dong & Yang (2020) and our reproductions show that GDAS is the best previous weight-sharing method; we evaluate GAEA GDAS and find that it achieves better results on CIFAR-100 and similar results on the other two datasets.

Table 2: **NAS-Bench-201:** Results are separated into traditional hyperparameter optimization algorithms with search run on CIFAR-10 (top block), weight-sharing methods with search run on CIFAR-10 (middle block), and weight-sharing methods run directly on the dataset used for training (bottom block). The use of transfer NAS follows the evaluations conducted by Dong & Yang (2020); unless otherwise stated all non-GAEA results are from their paper. The best results in the transfer and direct settings on each dataset are **bolded**.

| | | Search[*] (seconds) | CIFAR-10 (test) | CIFAR-100 (test) | ImageNet-16-120 (test) |
|---|---|---|---|---|---|
| Regular HO, search on CIFAR-10 | REA | N/A | $93.92 \pm 0.30$ | $71.84 \pm 0.99$ | $45.54 \pm 1.03$ |
| | RS | N/A | $93.70 \pm 0.36$ | $71.04 \pm 1.08$ | $44.57 \pm 1.25$ |
| | REINFORCE | N/A | $93.85 \pm 0.37$ | $71.71 \pm 1.09$ | $45.25 \pm 1.18$ |
| | BOHB | N/A | $93.61 \pm 0.52$ | $70.85 \pm 1.28$ | $44.42 \pm 1.49$ |
| Weight sharing, search on CIFAR-10 | RSPS | 7587 | $87.66 \pm 1.69$ | $58.33 \pm 4.34$ | $31.14 \pm 3.88$ |
| | DARTS (bilevel) | 35781 | $54.30 \pm 0.00$ | $15.61 \pm 0.00$ | $16.32 \pm 0.00$ |
| | SETN | 34139 | $87.64 \pm 0.00$ | $59.05 \pm 0.24$ | $32.52 \pm 0.21$ |
| | GDAS | 31609 | $93.61 \pm 0.09$ | $70.70 \pm 0.30$ | $41.71 \pm 0.98$ |
| | DARTS[‡] (bilevel) | $10683^\dagger$ | $54.30 \pm 0.00$ | $15.32 \pm 0.00$ | $16.38 \pm 0.00$ |
| | **GAEA** DARTS (bilevel) | $7930^\dagger$ | $91.63 \pm 2.57$ | $68.39 \pm 4.47$ | $41.59 \pm 4.20$ |
| | DARTS[‡] (ERM) | $18112^\dagger$ | $84.39 \pm 3.82$ | $54.81 \pm 7.08$ | $31.82 \pm 4.78$ |
| | **GAEA** DARTS (ERM) | $9061^\dagger$ | $\mathbf{94.10 \pm 0.29}$ | $\mathbf{72.60 \pm 0.89}$ | $\mathbf{45.81 \pm 0.51}$ |
| Weight sharing, direct search | GDAS[‡] | $27923^\dagger$ | $93.52 \pm 0.15$ | $67.52 \pm 0.15$ | $40.91 \pm 0.12$ |
| | **GAEA** GDAS | $16754^\dagger$ | $93.55 \pm 0.13$ | $70.47 \pm 0.47$ | $40.91 \pm 0.12$ |
| | DARTS[‡] (bilevel) | $10683^\dagger$ | $54.30 \pm 0.00$ | $15.32 \pm 0.00$ | $28.96 \pm 10.22$ |
| | **GAEA** DARTS (bilevel) | $7930^\dagger$ | $91.63 \pm 2.57$ | $71.87 \pm 0.57$ | $45.69 \pm 0.56$ |
| | DARTS[‡] (ERM) | $18112^\dagger$ | $84.39 \pm 3.82$ | $51.26 \pm 6.14$ | $31.35 \pm 7.46$ |
| | **GAEA** DARTS (ERM) | $9061^\dagger$ | $\mathbf{94.10 \pm 0.29}$ | $\mathbf{73.43 \pm 0.13}$ | $\mathbf{46.36 \pm 0.00}$ |
| | ResNet | N/A | 93.97 | 70.86 | 43.63 |
| | Optimal | N/A | 94.37 | 73.51 | 47.31 |

[*] Search cost reported for running the search algorithm on CIFAR-10.
[†] Search cost measured on NVIDIA P100 GPUs.
[‡] Our reproduction or implementation of a non-GAEA method.

Since we are interested in improving upon not only GAEA GDAS but also upon traditional hyperparameter optimization methods, we also investigate the performance of GAEA applied to first-order DARTS. We evaluate GAEA DARTS with both single-level (ERM) and bilevel optimization; recall that in the latter case we optimize architecture parameters w.r.t. the validation loss and the shared weights w.r.t. the training loss, whereas in ERM there is no data split. GAEA DARTS (ERM) achieves state-of-the-art performance on all three datasets in both the transfer and direct setting, exceeding the test accuracy of both weight-sharing and traditional hyperparameter tuning by a wide margin. GAEA DARTS (bilevel) performs worse but still exceeds all other methods on CIFAR-100 and ImageNet-16-120 in the direct search setting. The result thus also confirms the relevance of studying the single-level case to understand NAS; notably, the DARTS (ERM) baseline also improves substantially upon the DARTS (bilevel) baseline.

## 5 CONCLUSION

In this paper we take an optimization-based view of NAS, arguing that the design of good NAS algorithms is largely a matter of successfully optimizing and regularizing the supernet. In support of this, we develop GAEA, a simple modification of gradient-based NAS that attains state-of-the-art performance on several computer vision benchmarks while enjoying favorable speed and sparsity properties. We believe that obtaining high-performance NAS algorithms for a wide variety of applications will continue to require a similar co-design of search space parameterizations and optimization methods, and that our geometry-aware framework can help accelerate this process. In particular, most modern NAS algorithms search over products of categorical decision spaces, to which our approach is directly applicable. More generally, as the field moves towards more ambitious search spaces, e.g. full-network topologies or generalizations of operations such as convolution or attention, these developments may result in new architecture domains for which our work can inform the design of appropriate, geometry-aware optimization methods.

## ACKNOWLEDGMENTS

We thank Jeremy Cohen, Jeffrey Li, and Nicholas Roberts for helpful feedback. This work was supported in part by DARPA under cooperative agreements FA875017C0141 and HR0011202000, NSF grants CCF-1535967, CCF-1910321, IIS-1618714, IIS-1705121, IIS-1838017, IIS-1901403, and IIS-2046613, a Microsoft Research Faculty Fellowship, a Bloomberg Data Science research grant, an Amazon Research Award, an AWS Machine Learning Research Award, a Facebook Faculty Research Award, funding from Booz Allen Hamilton Inc., a Block Center Grant, a Carnegie Bosch Institute Research Award, and a Two Sigma Fellowship Award. Any opinions, findings and conclusions, or recommendations expressed in this material are those of the authors and do not necessarily reflect the views of DARPA, NSF, or any other funding agency.

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

## A  BACKGROUND ON NAS WITH WEIGHT-SHARING

Here we review the NAS setup motivating our work. Weight-sharing methods almost exclusively use micro cell-based search spaces for their tractability and additional structure (Pham et al., 2018; Liu et al., 2019). These search spaces can be represented as directed acyclic graphs (DAGs) with a set of ordered nodes $N$ and edges $E$. Each node $x^{(i)} \in N$ is a feature representation and each edge $(i, j) \in E$ is associated with an operation on the feature of node $j$ passed to node $i$ and aggregated with other inputs to form $x^{(j)}$, with the restriction that a given node $j$ can only receive edges from prior nodes as input. Hence, the feature at node $i$ is $x^{(i)} = \sum_{j<i} o^{(i,j)}(x^{(j)})$. Search spaces are specified by the number of nodes, the number of edges per node, and the set of operations $O$ that can be applied at each edge. Thus for NAS, $\mathcal{A} \subset \{0, 1\}^{|E| \times |O|}$ is the set of all valid architectures for encoded by edge and operation decisions. Treating both the *shared weights* $\mathbf{w} \in \mathbb{R}^d$ and architecture decisions $a \in \mathcal{A}$ as parameters, weight-sharing methods train a single network subsuming all possible functions within the search space.

Gradient-based weight-sharing methods apply continuous relaxations to the architecture space $\mathcal{A}$ in order to compute gradients in a continuous space $\Theta$. Methods like DARTS (Liu et al., 2019) and its variants (Chen et al., 2019; Laube & Zell, 2019; Hundt et al., 2019; Liang et al., 2019; Noy et al., 2019; Nayman et al., 2019) relax the search space by considering a *mixture* of operations per edge. For example, we will consider a relaxation where the architecture space $\mathcal{A} = \{0, 1\}^{|E| \times |O|}$ is relaxed into $\Theta = [0, 1]^{|E| \times |O|}$ with the constraint that $\sum_{o \in O} \theta_{i,o} = 1$, i.e. the operation weights on each edge sum to 1. The feature at node $i$ is then $x^{(i)} = \sum_{j<i} \sum_{o \in O} \theta_{i,j,o} o(x^{(j)})$. To get a valid architecture $a \in \mathcal{A}$ from a mixture $\theta$, rounding and pruning are typically employed after the search phase.

An alternative, *stochastic* approach, such as that used by GDAS (Dong & Yang, 2019), instead uses $\Theta$-parameterized distributions $p_\theta$ over $\mathcal{A}$ to sample architectures (Pham et al., 2018; Xie et al., 2019; Akimoto et al., 2019; Cai et al., 2019); unbiased gradients w.r.t. $\theta \in \Theta$ can be computed using Monte Carlo sampling. The goal of all these relaxations is to use simple gradient-based approaches to approximately optimize (1) over $a \in \mathcal{A}$ by optimizing (2) over $\theta \in \Theta$ instead. However, both the relaxation and the optimizer critically affect the convergence speed and solution quality. We next present a principled approach for understanding both mixture and stochastic methods.

# B  OPTIMIZATION

This section contains proofs and generalizations of the non-convex optimization results in Section 3. Throughout this section, $\mathbb{V}$ denotes a finite-dimensional real vector space with Euclidean inner product $\langle \cdot, \cdot \rangle$, $\mathbb{R}_+$ denotes the set of nonnegative real numbers, and $\overline{\mathbb{R}}$ denotes the set of extended real numbers $\mathbb{R} \cup \{\pm\infty\}$.

## B.1  PRELIMINARIES

**Definition 1.** *Consider a closed and convex subset $\mathcal{X} \subset \mathbb{V}$. For any $\alpha > 0$ and norm $\|\cdot\| : \mathcal{X} \mapsto \mathbb{R}_+$ an everywhere-subdifferentiable function $f : \mathcal{X} \mapsto \overline{\mathbb{R}}$ is called $\alpha$-**strongly-convex** w.r.t. $\|\cdot\|$ if $\forall\, \mathbf{x}, \mathbf{y} \in \mathcal{X}$ we have*

$$f(\mathbf{y}) \geq f(\mathbf{x}) + \langle \nabla f(\mathbf{x}), \mathbf{y} - \mathbf{x} \rangle + \frac{\alpha}{2} \|\mathbf{y} - \mathbf{x}\|^2$$

**Definition 2.** *Consider a closed and convex subset $\mathcal{X} \subset \mathbb{V}$. For any $\beta > 0$ and norm $\|\cdot\| : \mathcal{X} \mapsto \mathbb{R}_+$ an continuously-differentiable function $f : \mathcal{X} \mapsto \mathbb{R}$ is called $\beta$-**strongly-smooth** w.r.t. $\|\cdot\|$ if $\forall\, \mathbf{x}, \mathbf{y} \in \mathcal{X}$ we have*

$$f(\mathbf{y}) \leq f(\mathbf{x}) + \langle \nabla f(\mathbf{x}), \mathbf{y} - \mathbf{x} \rangle + \frac{\beta}{2} \|\mathbf{y} - \mathbf{x}\|^2$$

**Definition 3.** *Let $\mathcal{X}$ be a closed and convex subset of $\mathbb{V}$. The **Bregman divergence** induced by a strictly convex, continuously-differentiable **distance-generating function (DGF)** $\phi : \mathcal{X} \mapsto \overline{\mathbb{R}}$ is*

$$D_\phi(\mathbf{x}||\mathbf{y}) = \phi(\mathbf{x}) - \phi(\mathbf{y}) - \langle \nabla \phi(\mathbf{y}), \mathbf{x} - \mathbf{y} \rangle \; \forall\, \mathbf{x}, \mathbf{y} \in \mathcal{X}$$

*By definition, the Bregman divergence satisfies the following properties:*

1. *$D_\phi(\mathbf{x}||\mathbf{y}) \geq 0 \; \forall\, x, y \in \mathcal{X}$ and $D_\phi(\mathbf{x}||\mathbf{y}) = 0 \iff x = y$.*

2. *If $\phi$ is $\alpha$-strongly-convex w.r.t. norm $\|\cdot\|$ then so is $D_\phi(\cdot||\mathbf{y}) \; \forall\, \mathbf{y} \in \mathcal{X}$. Furthermore, $D_\phi(\mathbf{x}||\mathbf{y}) \geq \frac{\alpha}{2}\|\mathbf{x} - \mathbf{y}\|^2 \; \forall\, \mathbf{x}, \mathbf{y} \in \mathcal{X}$.*

3. *If $\phi$ is $\beta$-strongly-smooth w.r.t. norm $\|\cdot\|$ then so is $D_\phi(\cdot||\mathbf{y}) \; \forall\, \mathbf{y} \in \mathcal{X}$. Furthermore, $D_\phi(\mathbf{x}||\mathbf{y}) \leq \frac{\beta}{2}\|\mathbf{x} - \mathbf{y}\|^2 \; \forall\, \mathbf{x}, \mathbf{y} \in \mathcal{X}$.*

**Definition 4.** *(Zhang & He, 2018, Definition 2.1) Consider a closed and convex subset $\mathcal{X} \subset \mathbb{V}$. For any $\gamma > 0$ and $\phi : \mathcal{X} \mapsto \overline{\mathbb{R}}$ an everywhere-subdifferentiable function $f : \mathcal{X} \mapsto \mathbb{R}$ is called $\gamma$-**relatively-weakly-convex** ($\gamma$-**RWC**) w.r.t. $\phi$ if $f(\cdot) + \gamma\phi(\cdot)$ is convex on $\mathcal{X}$.*

**Definition 5.** *(Zhang & He, 2018, Definition 2.3) Consider a closed and convex subset $\mathcal{X} \subset \mathbb{V}$. For any $\lambda > 0$, function $f : \mathcal{X} \mapsto \mathbb{R}$, and DGF $\phi : \mathcal{X} \mapsto \overline{\mathbb{R}}$ the **Bregman proximal operator** of $f$ is*

$$\mathrm{prox}_\lambda(\mathbf{x}) = \underset{\mathbf{u} \in \mathcal{X}}{\arg\min}\; \lambda f(\mathbf{u}) + D_\phi(\mathbf{u}||\mathbf{x})$$

**Definition 6.** *(Zhang & He, 2018, Equation 2.11) Consider a closed and convex subset $\mathcal{X} \subset \mathbb{V}$. For any $\lambda > 0$, function $f : \mathcal{X} \mapsto \mathbb{R}$, and DGF $\phi : \mathcal{X} \mapsto \overline{\mathbb{R}}$ the **Bregman stationarity** of $f$ at any point $\mathbf{x} \in \mathcal{X}$ is*

$$\Delta_\lambda(\mathbf{x}) = \frac{D_\phi(\mathbf{x}|| \mathrm{prox}_\lambda(\mathbf{x})) + D_\phi(\mathrm{prox}_\lambda(\mathbf{x})||\mathbf{x})}{\lambda^2}$$

## B.2 RESULTS

Throughout this subsection let $\mathbb{V} = \bigtimes_{i=1}^{b} \mathbb{V}_i$ be a product space of $b$ finite-dimensional real vector spaces $\mathbb{V}_i$, each with an associated norm $\| \cdot \|_i : \mathbb{V}_i \mapsto \mathbb{R}_+$, and $\mathcal{X} = \bigtimes_{i=1}^{b} \mathcal{X}_i$ be a product set of $b$ subsets $\mathcal{X}_i \subset \mathbb{V}_i$, each with an associated 1-strongly-convex DGF $\phi_i : \mathcal{X}_i \mapsto \overline{\mathbb{R}}$ w.r.t. $\| \cdot \|_i$. For each $i \in [b]$ will use $\| \cdot \|_{i,*}$ to denote the dual norm of $\| \cdot \|_i$ and for any element $\mathbf{x} \in \mathcal{X}$ we will use $\mathbf{x}_i$ to denote its component in block $i$ and $\mathbf{x}_{-i}$ to denote the component across all blocks other than $i$. Define the functions $\| \cdot \| : \mathbb{V} \mapsto \mathbb{R}_+$ and $\| \cdot \|_* \mathbb{V} \mapsto \mathbb{R}_+$ for any $\mathbf{x} \in \mathbb{V}$ by $\|\mathbf{x}\|^2 = \sum_{i=1}^{b} \|\mathbf{x}_i\|_i^2$ and $\|\mathbf{x}\|_*^2 = \sum_{i=1}^{b} \|\mathbf{x}_i\|_{i,*}^2$, respectively, and the function $\phi : \mathcal{X} \mapsto \overline{\mathbb{R}}$ for any $\mathbf{x} \in \mathcal{X}$ by $\phi(\mathbf{x}) = \sum_{i=1}^{b} \phi_i(\mathbf{x})$. Finally, for any $n \in \mathbb{N}$ we will use $[n]$ to denote the set $\{1, \dots, n\}$.

**Setting 1.** *For some fixed constants $\gamma_i, L_i > 0$ for each $i \in [b]$ we have the following:*

1. *$f : \mathcal{X} \mapsto \mathbb{R}$ is everywhere-subdifferentiable with minimum $f^* > -\infty$ and for all $\mathbf{x} \in \mathcal{X}$ and each $i \in [b]$ the restriction $f(\cdot, \mathbf{x}_{-i})$ is $\gamma_i$-RWC w.r.t. $\phi_i$.*

2. *For each $i \in [b]$ there exists a stochastic oracle $G_i$ that for input $\mathbf{x} \in \mathcal{X}$ outputs a random vector $G_i(\mathbf{x}, \xi)$ s.t. $\mathbb{E}_\xi G_i(\mathbf{x}, \xi) \in \partial_i f(\mathbf{x})$, where $\partial_i f(\mathbf{x})$ is the subdifferential set of the restriction $f(\cdot, \mathbf{x}_{-i})$ at $\mathbf{x}_i$. Moreover, $\mathbb{E}_\xi \|G_i(\mathbf{x}, \xi)\|_{i,*}^2 \leq L_i^2$.*

*Define $\gamma = \max_{i \in [b]} \gamma_i$ and $L^2 = \sum_{i=1}^{b} L_i^2$.*

**Claim 1.** *$\| \cdot \|$ is a norm on $\mathbb{V}$.*

*Proof.* Positivity and homogeneity are trivial. For the triangle inequality, note that for any $\lambda \in [0, 1]$ and any $\mathbf{x}, \mathbf{y} \in \mathcal{X}$ we have that

$$
\begin{aligned}
\|\lambda \mathbf{x} + (1 - \lambda)\mathbf{y}\| &= \sqrt{\sum_{i=1}^{b} \|\lambda \mathbf{x}_i + (1 - \lambda)\mathbf{y}_i\|_i^2} \\
&\leq \sqrt{\sum_{i=1}^{b} (\lambda \|\mathbf{x}_i\|_i + (1 - \lambda)\|\mathbf{y}_i\|_i)^2} \\
&\leq \lambda \sqrt{\sum_{i=1}^{b} \|\mathbf{x}_i\|_i^2} + (1 - \lambda)\sqrt{\sum_{i=1}^{b} \|\mathbf{y}_i\|_i^2} \\
&= \lambda \|\mathbf{x}\| + (1 - \lambda)\|\mathbf{y}\|
\end{aligned}
$$

where the first inequality follows by convexity of the norms $\| \cdot \|_i \, \forall \, i \in [b]$ and the fact that the Euclidean norm on $\mathbb{R}^b$ is nondecreasing in each argument, while the second inequality follows by convexity of the Euclidean norm on $\mathbb{R}^b$. Setting $\lambda = \frac{1}{2}$ and multiplying both sides by 2 yields the triangle inequality. $\square$

---

**Algorithm 2:** Block-stochastic mirror descent over $\mathcal{X} = \bigtimes_{i=1}^{b} \mathcal{X}_i$ given associated DGFs $\phi_i : \mathcal{X}_i \mapsto \overline{\mathbb{R}}$.

---

**Input:** initialization $\mathbf{x}^{(1)} \in \mathcal{X}$, number of steps $T \geq 1$, step-size sequence $\{\eta_t\}_{t=1}^{T}$
**for** iteration $t \in [T]$ **do**
 $\quad$ sample $i \sim \mathrm{Unif}[b]$
 $\quad$ set $\mathbf{x}_{-i}^{(t+1)} = \mathbf{x}_{-i}^{(t)}$
 $\quad$ get $\mathbf{g} = G_i(\mathbf{x}^{(t)}, \xi_t)$
 $\quad$ set $\mathbf{x}_i^{(t+1)} = \arg\min_{\mathbf{u} \in \mathcal{X}_i} \eta_t \langle \mathbf{g}, \mathbf{u} \rangle + D_{\phi_i}(\mathbf{u} \| \mathbf{x}_i^{(t)})$
**Output:** $\hat{\mathbf{x}} = \mathbf{x}^{(t)}$ w.p. $\frac{\eta_t}{\sum_{t=1}^{T} \eta_t}$.

---

**Claim 2.** $\frac{1}{2} \| \cdot \|_*^2$ *is the convex conjugate of* $\frac{1}{2} \| \cdot \|^2$.

*Proof.* Consider any $\mathbf{u} \in \mathbb{V}$. To upper-bound the convex conjugate note that

$$
\sup_{\mathbf{x} \in \mathbb{V}} \langle \mathbf{u}, \mathbf{x} \rangle - \frac{\|\mathbf{x}\|^2}{2} = \sup_{\mathbf{x} \in \mathbb{V}} \sum_{i=1}^{b} \langle \mathbf{u}_i, \mathbf{x}_i \rangle - \frac{\|\mathbf{x}_i\|_i^2}{2}
$$

$$
\leq \sup_{\mathbf{x} \in \mathbb{V}} \sum_{i=1}^{b} \|\mathbf{u}_i\|_{i,*} \|\mathbf{x}_i\|_i - \frac{\|\mathbf{x}_i\|_i^2}{2}
$$

$$
= \frac{1}{2} \sum_{i=1}^{b} \|\mathbf{u}_i\|_{i,*}^2
$$

$$
= \frac{\|\mathbf{u}\|_*^2}{2}
$$

where the first inequality follows by definition of a dual norm and the second by maximizing each term w.r.t. $\|\mathbf{x}_i\|_i$. For the lower bound, pick $\mathbf{x} \in \mathbb{V}$ s.t. $\langle \mathbf{u}_i, \mathbf{x}_i \rangle = \|\mathbf{u}_i\|_{i,*} \|\mathbf{x}_i\|_i$ and $\|\mathbf{x}_i\|_i = \|\mathbf{u}_i\|_{i,*} \forall i \in [b]$, which must exist by the definition of a dual norm. Then

$$
\langle \mathbf{u}, \mathbf{x} \rangle - \frac{\|\mathbf{x}\|^2}{2} = \sum_{i=1}^{b} \langle \mathbf{u}_i, \mathbf{x}_i \rangle - \frac{\|\mathbf{x}_i\|_i^2}{2} = \frac{1}{2} \sum_{i=1}^{b} \frac{\|\mathbf{u}_i\|_{i,*}^2}{2} = \frac{\|\mathbf{u}\|_*^2}{2}
$$

so $\sup_{\mathbf{x} \in \mathbb{V}} \langle \mathbf{u}, \mathbf{x} \rangle - \frac{1}{2} \|\mathbf{x}\|^2 \geq \frac{1}{2} \|\mathbf{u}\|_*^2$, completing the proof. $\qquad \square$

**Theorem 2.** *Let $\hat{x}$ be the output of Algorithm 2 after $T$ iterations with non-increasing step-size sequence $\{\eta_t\}_{t=1}^{T}$. Then under Setting 1, for any $\hat{\gamma} > \gamma$ we have that*

$$\mathbb{E}\Delta_{\frac{1}{\hat{\gamma}}}(\hat{\mathbf{x}}) \leq \frac{\hat{\gamma}b}{\hat{\gamma} - \gamma} \frac{\min_{\mathbf{u}\in\mathcal{X}} f(\mathbf{u}) + \hat{\gamma}D_{\phi}(\mathbf{u}||\mathbf{x}^{(1)}) - f^* + \frac{\hat{\gamma}L^2}{2b}\sum_{t=1}^{T}\eta_t^2}{\sum_{t=1}^{T}\eta_t}$$

*where the expectation is w.r.t. $\xi_t$ and the randomness of the algorithm.*

*Proof.* Define transforms $\mathbf{U}_i, i \in [b]$ s.t. $\mathbf{U}_i^T\mathbf{x} = \mathbf{x}_i$ and $\mathbf{x} = \sum_{i=1}^{b}\mathbf{U}_i\mathbf{x}_i \ \forall \ \mathbf{x} \in \mathcal{X}$. Let $G$ be a stochastic oracle that for input $\mathbf{x} \in \mathcal{X}$ outputs $G(\mathbf{x}, i, \xi) = b\mathbf{U}_iG_i(\mathbf{x}, \xi)$. This implies $\mathbb{E}_{i,\xi}G(\mathbf{x}, i, \xi) = \frac{1}{b}\sum_{i=1}^{b}b\mathbf{U}_i\mathbb{E}_{\xi}G_i(\mathbf{x}, \xi) \in \sum_{i=1}^{b}\mathbf{U}_i\partial_i f(\mathbf{x}) = \partial f(\mathbf{x})$ and $\mathbb{E}_{i,\xi}||G(\mathbf{x}, i, \xi)||_*^2 = \frac{1}{b}\sum_{i=1}^{b}b^2\mathbb{E}_{\xi}||\mathbf{U}_iG_i(\mathbf{x}, \xi)||_{i,*}^2 \leq b\sum_{i=1}^{b}L_i^2 = bL^2$. Then

$$\mathbf{x}^{(t+1)} = \mathbf{U}_i\left[\arg\min_{\mathbf{u}\in\mathcal{X}_i}\eta_t\langle g, u\rangle + D_{\phi_i}(\mathbf{u}||\mathbf{x}_i^{(t)})\right]$$

$$= \mathbf{U}_i\mathbf{U}_i^T\left[\arg\min_{\mathbf{u}\in\mathcal{X}}\eta_t\langle\mathbf{U}_iG_i(\mathbf{x}^{(t)}, \xi_t), \mathbf{u}\rangle + \sum_{i=1}^{b}D_{\phi_i}(\mathbf{u}_i||\mathbf{x}_i^{(t)})\right]$$

$$= \arg\min_{\mathbf{u}\in\mathcal{X}}\frac{\eta_t}{b}\langle G(\mathbf{x}, i, \xi_t), \mathbf{u}\rangle + D_{\phi}(\mathbf{u}||\mathbf{x}^{(t)})$$

Thus Algorithm 2 is equivalent to Zhang & He (2018, Algorithm 1) with stochastic oracle $G(\mathbf{x}, i, \xi)$, step-size sequence $\{\eta_t/b\}_{t=1}^{T}$, and no regularizer. Note that $\phi$ is 1-strongly-convex w.r.t. $||\cdot||$ and $f$ is $\gamma$-RWC w.r.t. $\phi$, so in light of Claims 1 and 2 our setup satisfies Assumption 3.1 of Zhang & He (2018). The result then follows from Theorem 3.1 of the same. □

**Corollary 1.** *Under Setting 1 let $\hat{\mathbf{x}}$ be the output of Algorithm 2 with constant step-size $\eta_t = \sqrt{\frac{2b(f^{(1)}-f^*)}{\gamma L^2 T}} \ \forall \ t \in [T]$, where $f^{(1)} = f(\mathbf{x}^{(1)})$. Then we have*

$$\mathbb{E}\Delta_{\frac{1}{2\gamma}}(\hat{\mathbf{x}}) \leq 2L\sqrt{\frac{2b\gamma(f^{(1)} - f^*)}{T}}$$

*where the expectation is w.r.t. $\xi_t$ and the randomness of the algorithm. Equivalently, we can reach a point $\hat{\mathbf{x}}$ satisfying $\mathbb{E}\Delta_{\frac{1}{2\gamma}}(\hat{\mathbf{x}}) \leq \varepsilon$ in $\frac{8\gamma bL^2(f^{(1)}-f^*)}{\varepsilon^2}$ stochastic oracle calls.*

### B.3 A Single-Level Analysis of ENAS and DARTS

In this section we apply our analysis to understanding two existing NAS algorithms, ENAS (Pham et al., 2018) and DARTS (Liu et al., 2019). For simplicity, we assume objectives induced by architectures in the relaxed search space are $\gamma$-smooth, which excludes components such as ReLU. However, such cases can be smoothed via Gaussian convolution, i.e. adding noise to every gradient; thus given the noisiness of SGD training we believe the following analysis is still informative (Kleinberg et al., 2018).

ENAS continuously relaxes $\mathcal{A}$ via a neural controller that samples architectures $a \in \mathcal{A}$, so $\Theta = \mathbb{R}^{\mathcal{O}(h^2)}$, where $h$ is the number of hidden units. The controller is trained with Monte Carlo gradients. On the other hand, first-order DARTS uses a mixture relaxation similar to the one in Section A but using a softmax instead of constraining parameters to the simplex. Thus $\Theta = \mathbb{R}^{|E| \times |O|}$ for $E$ the set of learnable edges and $O$ the set of possible operations. If we assume that both algorithms use SGD for the architecture parameters then to compare them we are interested in their respective values of $G_\theta$, which we will refer to as $G_{\text{ENAS}}$ and $G_{\text{DARTS}}$. Before proceeding, we note again that our theory holds only for the single-level objective and when using SGD as the architecture optimizer, whereas both algorithms specify the bilevel objective and Adam (Kingma & Ba, 2015), respectively.

At a very high level, the Monte Carlo gradients used by ENAS are known to be high-variance, so $G_{\text{ENAS}}$ may be much larger than $G_{\text{DARTS}}$, yielding faster convergence for DARTS, which is reflected in practice (Liu et al., 2019). We can also do a simple low-level analysis under the assumption that all architecture gradients are bounded entry-wise, i.e. in $\ell_\infty$-norm, by some constant; then since the squared $\ell_2$-norm is bounded by the product of the dimension and the squared $\ell_\infty$-norm we have $G_{\text{ENAS}}^2 = \mathcal{O}(h^2)$ while $G_{\text{DARTS}}^2 = \mathcal{O}(|E||O|)$. Since ENAS uses a hidden state size of $h = 100$ and the DARTS search space has $|E| = 14$ edges and $|O| = 7$ operations, this also points to DARTS needing fewer iterations to converge.

## C  EXPERIMENTAL DETAILS

We provide additional detail on the experimental setup and hyperparameter settings used for each benchmark studied in Section 4. We also provide a more detailed discussion of how XNAS differs from GAEA, along with empirical results for XNAS on the NAS-Bench-201 benchmark.

### C.1  DARTS SEARCH SPACE

We consider the same search space as DARTS (Liu et al., 2019), which has become one of the standard search spaces for CNN cell search (Xie et al., 2019; Nayman et al., 2019; Chen et al., 2019; Noy et al., 2019; Liang et al., 2019). Following the evaluation procedure used in Liu et al. (2019) and Xu et al. (2020) , our evaluation of GAEA PC-DARTS consists of three stages:

- **Stage 1:** In the search phase, we run GAEA PC-DARTS with 5 random seeds to reduce variance from different initialization of the shared-weights network.[3]
- **Stage 2:** We evaluate the best architecture identified by each search run by training from scratch.
- **Stage 3:** We perform a more thorough evaluation of the best architecture from stage 2 by training with ten different random seed initializations.

For completeness, we describe the convolutional neural network search space considered. A cell consists of 2 input nodes and 4 intermediate nodes for a total of 6 nodes. The nodes are ordered and subsequent nodes can receive the output of prior nodes as input so for a given node $k$, there are $k-1$ possible input edges to node $k$. Therefore, there are a total of $2+3+4+5=14$ edges in the weight-sharing network.

An architecture is defined by selecting 2 input edges per intermediate node and also selecting a single operation per edge from the following 8 operations: (1) $3 \times 3$ separable convolution, (2) $5 \times 5$ separable convolution, (3) $3 \times 3$ dilated convolution, (4) $5 \times 5$ dilated convolution, (5) max pooling, (6) average pooling, (7) identity (8) zero. We use the same search space to design a "normal" cell and a "reduction" cell; the normal cells have stride 1 operations that do not change the dimension of the input, while the reduction cells have stride 2 operations that half the length and width dimensions of the input. In the experiments, for both cell types, , after which the output of all intermediate nodes are concatenated to form the output of the cell.

#### C.1.1  STAGE 1: ARCHITECTURE SEARCH

For stage 1, as is done by DARTS and PC-DARTS, we use GAEA PC-DARTS to update architecture parameters for a smaller shared-weights network in the search phase with 8 layers and 16 initial channels. All hyperparameters for training the weight-sharing network are the same as that used by PC-DARTS:

```
train:
    scheduler: cosine
    lr_anneal_cycles: 1
    smooth_cross_entropy: false
    batch_size: 256
    learning_rate: 0.1
    learning_rate_min: 0.0
    momentum: 0.9
    weight_decay: 0.0003
    init_channels: 16
    layers: 8
    autoaugment: false
    cutout: false
    auxiliary: false
    drop_path_prob: 0
    grad_clip: 5
```

[3]Note (Liu et al., 2019) trains the weight-sharing network with 4 random seeds. However, since PC-DARTS is significantly faster than DARTS, the cost of an additional seed is negligible.

For GAEA PC-DARTS, we initialize the architecture parameters with equal weight across all options (equal weight across all operations per edge and equal weight across all input edges per node). Then, we train the shared-weights network for 10 epochs without performing any architecture updates to warmup the weights. Then, we use a learning rate of 0.1 for the exponentiated gradient update for GAEA PC-DARTS.

### C.1.2 STAGE 2 AND 3: ARCHITECTURE EVALUATION

For stages 2 and 3, we train each architecture for 600 epochs with the same hyperparameter settings as PC-DARTS:

```
train:
    scheduler: cosine
    lr_anneal_cycles: 1
    smooth_cross_entropy: false
    batch_size: 128
    learning_rate: 0.025
    learning_rate_min: 0.0
    momentum: 0.9
    weight_decay: 0.0003
    init_channels: 36
    layers: 20
    autoaugment: false
    cutout: true
    cutout_length: 16
    auxiliary: true
    auxiliary_weight: 0.4
    drop_path_prob: 0.3
    grad_clip: 5
```

### C.1.3 BROAD REPRODUCIBILITY

Our 'broad reproducibility' results in Table 3 show the final stage 3 evaluation performance of GAEA PC-DARTS for 2 additional sets of random seeds from stage 1 search. The performance of GAEA PC-DARTS for one set is similar to that reported in Table 1, while the other is on par with the performance reported for PC-DARTS in Xu et al. (2020). We do observe non-negligible variance in the performance of the architecture found by different random seed initializations of the shared-weights network, necessitating running multiple searches before selecting an architecture. We also found that it was possible to identify and eliminate poor performing architectures in just 20 epochs of training during stage 2 intermediate evaluation, thereby reducing the total training cost by over 75% (we only trained 3 out of 10 architectures for the entire 600 epochs).

| Stage 3 Evaluation | | |
|---|---|---|
| Set 1 (Reported) | Set 2 | Set 3 |
| $2.50 \pm 0.07$ | $2.50 \pm 0.09$ | $2.60 \pm 0.09$ |

Table 3: GAEA PC-DARTS Stage 3 Evaluation for 3 sets of random seeds.

We depict the top architectures found by GAEA PC-DARTS for CIFAR-10 and ImageNet in Figure 3 and detailed results in Tables 4 and 5.

| Architecture | Test Error Best | Test Error Average | Params (M) | Search Cost (GPU Days) | Comparable Search Space | Search Method |
|---|---|---|---|---|---|---|
| Shake-Shake (DeVries & Taylor, 2017) | N/A | 2.56 | 26.2 | - | - | manual |
| PyramidNet (Yamada et al., 2018) | 2.31 | N/A | 26 | - | - | manual |
| NASNet-A* (Zoph et al., 2018) | N/A | 2.65 | 3.3 | 2000 | N | RL |
| AmoebaNet-B* (Real et al., 2019) | N/A | 2.55 ± 0.05 | 2.8 | 3150 | N | evolution |
| ProxylessNAS‡ (Cai et al., 2019) | 2.08 | N/A | 5.7 | 4 | N | gradient |
| ENAS (Pham et al., 2018) | 2.89 | N/A | 4.6 | 0.5 | Y | RL |
| Random search WS† (Li & Talwalkar, 2019) | 2.71 | 2.85 ± 0.08 | 3.8 | 0.7 | Y | random |
| ASNG-NAS (Akimoto et al., 2019) | N/A | 2.83 ± 0.14 | 3.9 | 0.1 | Y | gradient |
| SNAS (Xie et al., 2019) | N/A | 2.85 ± 0.02 | 2.8 | 1.5 | Y | gradient |
| DARTS (1st-order)† (Liu et al., 2019) | N/A | 3.00 ± 0.14 | 3.3 | 0.4 | Y | gradient |
| DARTS (2nd-order)† (Liu et al., 2019) | N/A | 2.76 ± 0.09 | 3.3 | 1 | Y | gradient |
| PDARTS# (Chen et al., 2019) | 2.50 | N/A | 3.4 | 0.3 | Y | gradient |
| PC-DARTS† (Xu et al., 2020) | N/A | 2.57 ± 0.07 | 3.6 | 0.1 | Y | gradient |
| **GAEA** PC-DARTS† (Ours) | 2.39 | 2.50 ± 0.06 | 3.7 | 0.1 | Y | gradient |

[*] We show results for networks with a comparable number of parameters.
[†] For fair comparison to other work, we show the search cost for training the shared-weights network with a single initialization.
[‡] Search space and backbone architecture (PyramidNet) differ from the DARTS setting.
[#] PDARTS results not reported for multiple seeds. Additionally, PDARTS uses deeper weight-sharing networks during search, on which PC-DARTS has also been shown to improve performance (Xu et al., 2020), so we GAEA PC-DARTS to further improve if modified similarly.

Table 4: **DARTS (CIFAR-10):** Comparison with manually designed networks and those found by SOTA NAS methods, mainly on the DARTS search space (Liu et al., 2019). Results grouped by the type of search method: manually designed, full-evaluation NAS, and weight-sharing NAS. All test errors are for models trained with auxiliary towers and cutout (parameter counts exclude auxiliary weights). Test errors we report are averaged over 10 seeds. "-" indicates that the field does not apply while "N/A" indicates unknown. Note that search cost is hardware-dependent; our results used Tesla V100 GPUs.

| Architecture | Source | Test Error top-1 | Test Error top-5 | Params (M) | Search Cost (GPU Days) | Search Method |
|---|---|---|---|---|---|---|
| MobileNet | (Howard et al., 2017) | 29.4 | 10.5 | 4.2 | - | manual |
| ShuffleNet V2 2x | (Ma et al., 2018) | 25.1 | N/A | ∼ 5 | - | manual |
| NASNet-A* | (Zoph et al., 2018) | 26.0 | 8.4 | 5.3 | 1800 | RL |
| AmoebaNet-C* | (Real et al., 2019) | 24.3 | 7.6 | 6.4 | 3150 | evolution |
| DARTS† | (Liu et al., 2019) | 26.7 | 8.7 | 4.7 | 4.0 | gradient |
| SNAS | (Xie et al., 2019) | 27.3 | 9.2 | 4.3 | 1.5 | gradient |
| ProxylessNAS‡ | (Cai et al., 2019) | 24.9 | 7.5 | 7.1 | 8.3 | gradient |
| PDARTS# | (Chen et al., 2019) | 24.4 | 7.4 | 4.9 | 0.3 | gradient |
| PC-DARTS (CIFAR-10)† | (Xu et al., 2020) | 25.1 | 7.8 | 5.3 | 0.1 | gradient |
| PC-DARTS (ImageNet)† | (Xu et al., 2020) | 24.2 | 7.3 | 5.3 | 3.8 | gradient |
| **GAEA** PC-DARTS (CIFAR-10)† | Ours | 24.3 | 7.3 | 5.3 | 0.1 | gradient |
| **GAEA** PC-DARTS (ImageNet)† | Ours | 24.0 | 7.3 | 5.6 | 3.8 | gradient |

[*],[†],[‡],[#] See notes in Table 4.

Table 5: **DARTS (ImageNet):** Comparison with manually designed networks and those found by SOTA NAS methods, mainly on the DARTS search space (Liu et al., 2019). Results are grouped by the type of search method: manually designed, full-evaluation NAS, and weight-sharing NAS. All test errors are for models trained with auxiliary towers and cutout but no other modifications, e.g. label smoothing (Müller et al., 2019), AutoAugment (Cubuk et al., 2019), Swish (Ramachandran et al., 2017), squeeze and excite modules (Hu et al., 2018), etc. "-" indicates that the field does not apply while "N/A" indicates unknown. Note that search cost is hardware-dependent; our results used Tesla V100 GPUs.

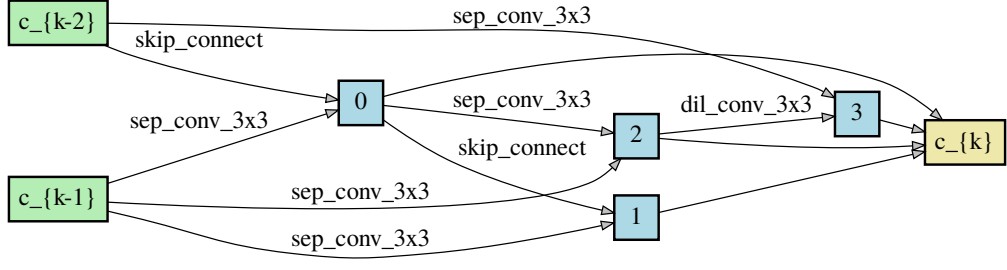

(a) CIFAR-10: Normal Cell

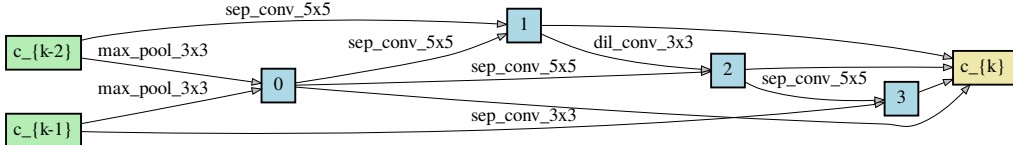

(b) CIFAR-10: Reduction Cell

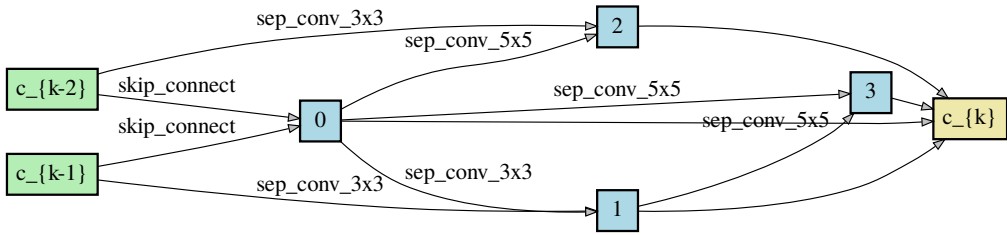

(c) ImageNet: Normal Cell

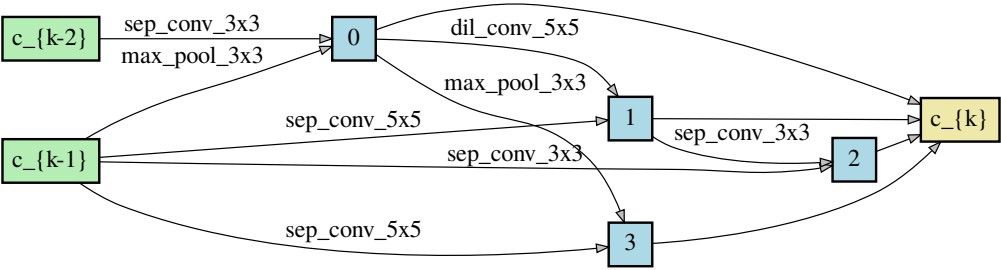

(d) ImageNet: Reduction Cell

Figure 3: The best normal and reduction cells found by GAEA PC-DARTS on CIFAR-10 (top) and ImageNet (bottom).

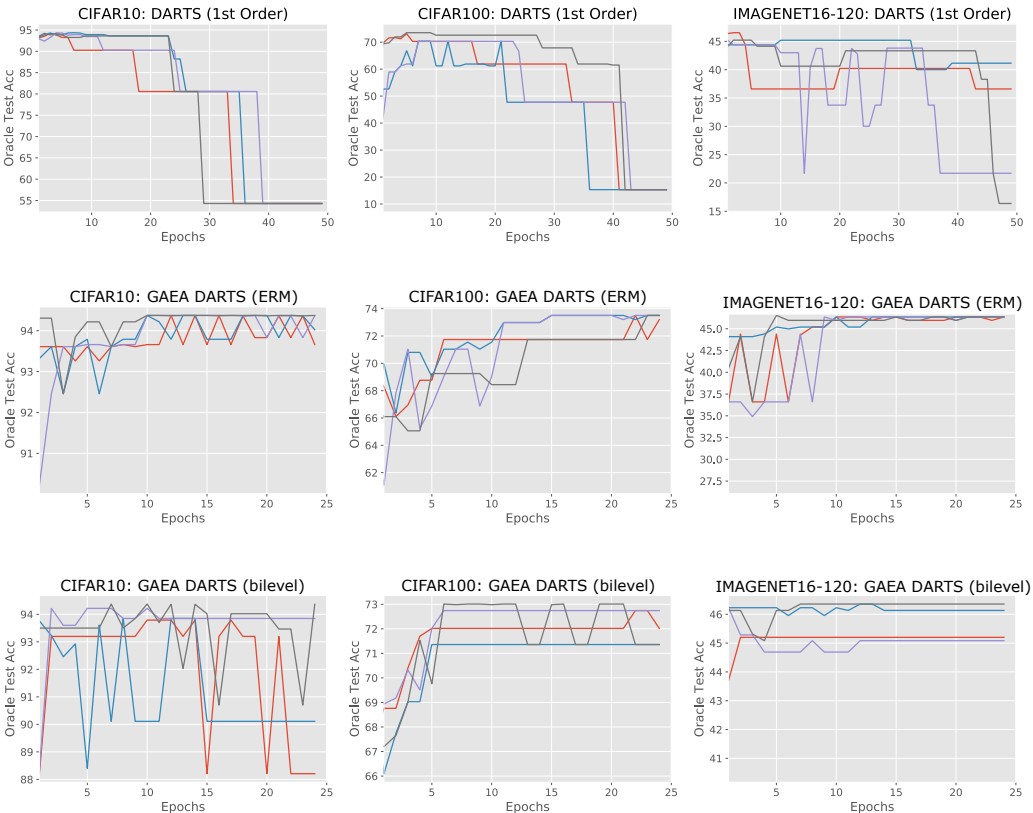

Figure 4: **NAS-Bench-201: Learning Curves.** Evolution over search phase epochs of the best architecture according to the NAS method. DARTS (first-order) converges to nearly all skip connections while GAEA is able to suppress overfitting to the mixture relaxation by encouraging sparsity in operation weights.

## C.2  NAS-BENCH-1SHOT1

The NAS-Bench-1Shot1 benchmark (Zela et al., 2020a) contains 3 different search spaces that are subsets of the NAS-Bench-101 search space. The search spaces differ in the number of nodes and the number of input edges selected per node. We refer the reader to (Zela et al., 2020a) for details about each individual search space.

Of the NAS methods evaluated in Zela et al. (2020a), PC-DARTS had the most robust performance across the three search spaces and converged to the best architecture in search spaces 1 and 3. GDAS, a probabilistic gradient NAS method, achieved the best performance on search space 2. Hence, we focused on applying a geometry-aware approach to PC-DARTS. We implemented GAEA PC-DARTS within the repository provided by the authors of Zela et al. (2020a) available at `https://github.com/automl/nasbench-1shot1`. We used the same hyperparameter settings for training the weight-sharing network as that used by Zela et al. (2020a) for PC-DARTS. Similar to the previous benchmark, we initialize architecture parameters to allocate equal weight to all options. For the architecture updates, the only hyperparameter for GAEA PC-DARTS is the learning rate for exponentiated gradient, which we set to 0.1.

As mentioned in Section 4, the search spaces considered in this benchmark differ in that operations are applied after aggregating all edge inputs to a node instead of per edge input as in the DARTS and NAS-Bench-201 search spaces. This structure inherently limits the size of the weight-sharing network to scale with the number of nodes instead of the number of edges ($\mathcal{O}(|\text{nodes}|^2)$), thereby limiting the degree of overparameterization. Understanding the impact of overparameterization on the performance of weight-sharing NAS methods is a direction for future study.

## C.3 NAS-BENCH-201

The NAS-Bench-201 benchmark (Dong & Yang, 2020) evaluates a single search space across 3 datasets: CIFAR-10, CIFAR-100, and a miniature version of ImageNet (ImageNet-16-120). ImageNet-16-120 is a downsampled version of ImageNet with $16 \times 16$ images and 120 classes for a total of 151.7k training images, 3k validation images, and 3k test images. The authors of Dong & Yang (2020) evaluated the architecture search performance of multiple weight-sharing methods and traditional hyperparameter optimization methods on all three datasets. According to the results from Dong & Yang (2020), GDAS outperformed other weight-sharing methods by a large margin. Hence, we first evaluated the performance of GAEA GDAS on each of the three datasets. Our implementation of GAEA GDAS uses an architecture learning rate of 0.1, which matches the learning rate used for GAEA approaches in the previous two benchmarks. Additionally, we run GAEA GDAS for 150 epochs instead of 250 epochs used for GDAS in the original benchmarked results; this is why the search cost is lower for GAEA GDAS. All other hyperparameter settings are the same. Our results for GAEA GDAS is comparable to the reported results for GDAS on CIFAR-10 and CIFAR-100 but slightly lower on ImageNet-16-120. Compared to our reproduced results for GDAS, GAEA GDAS outperforms GDAS on CIFAR-100 and matches it on CIFAR-10 and ImageNet-16-120.

Next, to see if we can use GAEA to further improve the performance of weight-sharing methods, we evaluated GAEA DARTS (first order) applied to both the single-level (ERM) and bi-level optimization problems. Again, we used a learning rate of 0.1 and trained GAEA DARTS for 25 epochs on each dataset. The one additional modification we made was to exclude the `zero` operation, which limits GAEA DARTS to a subset of the search space. To isolate the impact of this modification, we also evaluated first-order DARTS with this modification. Similar to (Dong & Yang, 2020), we observe DARTS with this modification to also converge to architectures with nearly all skip connections, resulting in similar performance as that reported in Dong & Yang (2020). We present the learning curves of the oracle architecture recommended by DARTS and GAEA DARTS (when excluding zero operation) over the training horizon for 4 different runs in Figure 4. For GAEA GDAS and GAEA DARTS, we train the weight-sharing network with the following hyperparameters:

```
train:
    scheduler: cosine
    lr_anneal_cycles: 1
    smooth_cross_entropy: false
    batch_size: 64
    learning_rate: 0.025
    learning_rate_min: 0.001
    momentum: 0.9
    weight_decay: 0.0003
    init_channels: 16
    layers: 5
    autoaugment: false
    cutout: false
    auxiliary: false
    auxiliary_weight: 0.4
    drop_path_prob: 0
    grad_clip: 5
```

Surprisingly, we observe single-level optimization to yield better performance than solving the bi-level problem with GAEA DARTS on this search space. In fact, the performance of GAEA DARTS (ERM) not only exceeds that of GDAS, but also outperforms traditional hyperparameter optimization approaches on all three datasets, nearly reaching the optimal accuracy on all three datasets. In contrast, GAEA DARTS (bi-level) outperforms GDAS on CIFAR-100 and ImageNet-16-120 but underperforms slightly on CIFAR-10. The single-level results on this benchmark provides concrete support for our convergence analysis, which only applies to the ERM problem. As noted in Section 4, the search space considered in this benchmark differs from the prior two in that there is no subsequent edge pruning. Additionally, the search space is fairly small with only 3 nodes for which architecture decisions must be made. The success of GAEA DARTS (ERM) on this benchmark indicate the need for a better understanding of when single-level optimization should be used in favor of the default bi-level optimization problem and how the search space impacts the decision.

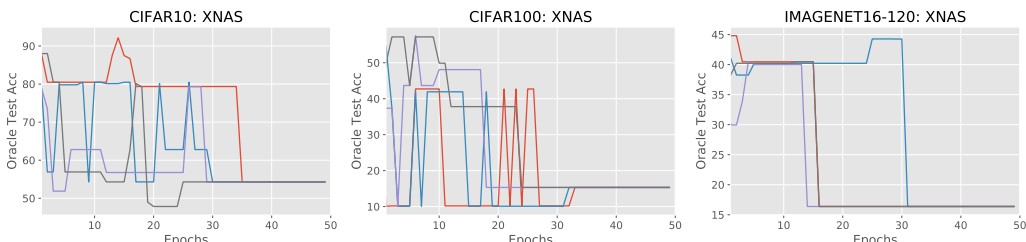

Figure 5: **NAS-Bench-201: XNAS Learning Curves.** Evolution over search phase epochs of the best architecture according 4 runs of XNAS. XNAS exhibits the same behavior as DARTS and converges to nearly all skip connections.

## C.4 COMPARISON TO XNAS

As discussed in Section 1, XNAS is similar to GAEA in that it uses an exponentiated gradient update but is motivated from a regret minimization perspective. Nayman et al. (2019) provides regret bounds for XNAS relative to the observed sequence of validation losses, however, this is not equivalent to the regret relative to the best architecture in the search space, which would have generated a different sequence of validation losses.

XNAS also differs in its implementation in two ways: (1) a wipeout routine is used to zero out operations that cannot recover to exceed the current best operation within the remaining number of iterations and (2) architecture gradient clipping is applied per data point before aggregating to form the update. These differences are motivated from the regret analysis and meaningfully increase the complexity of the algorithm. Unfortunately, the authors do not provide the code for architecture search in their code release at `https://github.com/NivNayman/XNAS`. Nonetheless, we implemented XNAS for the NAS-Bench-201 search space to provide a point of comparison to GAEA.

Our results shown in Figure 5 demonstrate that XNAS exhibits much of the same behavior as DARTS in that the operations all converge to skip connections. We hypothesize that this is due to the gradient clipping, which obscures the signal kept by GAEA in favor of convolutional operations.

