# OpenReview forum: "Geometry-Aware Gradient Algorithms for Neural Architecture Search"
_ICLR.cc/2021/Conference — ICLR 2021 Spotlight_

### Official Review · AnonReviewer1 · 2020-10-15
**Paper Review**

**Rating:** 7
**Confidence:** 3

**Review:**

## Summary
The submission presents a modification to the DARTS family of efficient Neural Architecture Search algorithms. The authors claim their modification (i) leads to better empirical performance, and (ii) is theoretically well-motivated. DARTS is a Neural Architecture Search algorithm which aims to find the most accurate network architecture within a human-defined search space.

The original DARTS algorithm uses gradient descent to optimize both (i) a set of shared model weights $\mathbf{w}$ which are used to rank different candidate architectures within a search space and (ii) a vector $\theta \in \mathbf{R}^n$ which corresponds to a continuous (real-valued) relaxation the network architecture. At the end of the search, $\theta$ is converted into a discrete architecture within the search space; this architecture is the output of the search process. The authors' proposed variant of DARTS -- which they call GAEA (Geometry-Aware Gradient Algorithm) -- updates the architectural parameters $\theta$ using *exponentiated* gradient descent instead of standard gradient descent. (Details are provided in Equation (8) in Section 3.3 of their paper, but the modification seems straightforward to implement.) The authors claim this modification favors sparse values of $\theta$, and helps decrease the gap between discrete architectures and their continuous relaxations.

On the applied side: The authors evaluate efficient NAS algorithms with and without their modification on three different benchmark tasks. On two of the tasks, the performance of their modified algorithm seems quite close to that of a baseline without the modification. On the third task (NASBench-201), the improvements appear to be quite substantial in one case. In most (but not all) cases, the authors directly compare the results of architecture searches obtained with and without their proposed modification. This type of control experiment is important and very helpful for a NAS paper, since it makes it possible to directly quantify the helpfulness of the proposed changes.

On the theoretical side: I had trouble understanding the significance of the authors' theoretical results. In particular: the paper claims to prove stationary-point convergence bounds for a variant of their algorithm but doesn't explain what stationary-point convergence means or why it's important. I'm very much on the applied side of machine learning, so it's possible that someone with a stronger theory background would get more out of the theoretical results. However, I also think the theoretical results could be much better-explained than they currently are. (I read through Section 3, but have not tried to check the theoretical results carefully.)

*Pros:*
* Paper provides experiments on three different benchmark tasks, and reports results for multiple NAS algorithms (mostly based on DARTS).
* Strong empirical results for some of the NASBench-201 experiments.
* Paper appears to provide solid baselines for most experiments.
* Proposed approach is not overly complex, and seems straightforward to implement.

*Cons:*
* Novelty is limited, since the authors are proposing a relatively limited change to an existing family of Efficient NAS algorithms. (This would, however, be a non-issue if the proposed changes were effective enough.)
* Unclear presentation of theoretical results.
* Quality improvements seem quite limited on 2 of the 3 benchmark tasks. (Part of the problem might -- as the authors suggest -- be due to limited headroom.)
* For the benchmark task with the largest improvement -- NASBench-201 -- the authors only provide their own baseline for 1 of the 3 NAS algorithms they evaluate, and for this variant (GDAS), the improvements over the baseline are quite small. So it's not clear to me how much of the improvements for the other two algorithms can be attributed to GAEA and how much can be attributed to NAS setups that are better tuned than the ones from the original NASBench-201 paper.

## Notes on Experimental Evaluation
The abstract makes a strong claim about the quality improvements brought about by the authors' proposed modification, so I'll evaluate the paper relative to that claim: "we exceed the best published results for both CIFAR and ImageNet on both the DARTS search space and NAS-Bench-201; on the latter we achieve near-oracle-optimal performance on CIFAR-10 and CIFAR-100."

The submission evaluates their method on three tasks: (i) the search space from the DARTS paper, (ii) the NASBench-1shot1 benchmark task, and (iii) the NASBench-201 benchmark task. On the plus side: the authors appear to provide direct comparisons between DARTS-based NAS algorithms with and without their proposed GAEA modifications (although this requires some confirmation of details which are currently missing from the paper). On the minus side: the improvements of GAEA algorithms over their non-GAEA equivalents seem quite small in most cases. (The authors attribute this to limited headroom in the search spaces, which seems plausible to me.)

On the original DARTS search space, the authors' proposed GAEA PC-DARTS algorithm produces architectures with an average test error rate of 2.50%, compared with 2.57% for standard PC-DARTS. The authors do not provide numbers for their own reproduction of the original PC-DARTS results; the writing implies that the hyper-parameters and search space are comparable. Providing numbers for the authors' reproduction of PC-DARTS would strengthen these results. Assuming the current numbers are in fact comparable: the performance improvements appear to be extremely small, and are probably within the range of statistical error.

On the NASBench-1shot1 search space, the authors claim that their proposed GAEA PC-DARTS algorithm works slightly better than an unmodified PC-DARTS, although the improvements appear to be small. While this seems reasonable given the data they provided, I think the presentation of results can be improved; the results presented in Figure 2 are difficult for me to interpret because the plots are visually noisy and contain many overlapping lines of different colors.

NASBench-201 is the most interesting (and potentially most appealing) case. The authors provide accuracy numbers for both their reproduction of the existing GDAS algorithm and their proposed GAEA-GDAS variant, and performance of the two seems quite similar (e.g., 93.55% test accuracy on CIFAR-10 for GDAS-GAEA compared with 93.52% for GDAS). The gap is larger on CIFAR-100, but primarily because the authors' reproduction of GDAS has lower accuracy than the published GDAS numbers from the NASBench-201 paper. For other DARTS variants, the accuracy improvements are quite impressive (e.g., 94.10% accuracy for GAEA-DARTS-ERM), but the authors do not provide a non-GAEA baseline for this configuration. Furthermore, the accuracies that the original NASBench-201 paper provides for baselines such as ENAS and DARTS are extremely low, so I think that providing a careful reproduction of DARTS-ERM with properly tuned hyper-parameters would help make the results more convincing.

## Notes on Clarity
While the paper seems reasonably well-organized and well-proofread, I found parts of it difficult to understand. I had trouble understanding Section 3 (which presents the paper's main theoretical results) in particular. The section is notation and terminology-heavy, and would be easier to follow if it spent more time explaining relevant terms.

I found Table 1 difficult to interpret because different rows of the table contain the results of architecture searches performed using different search spaces and/or model training hyper-parameters. For example, ProxylessNAS uses a very different search space than PC-DARTS and GAEA PC-DARTS. While the authors note this particular example in the text under the table, the table would be much easier to interpret if it was organized to show which subsets of the results were comparable to each other (i.e., used the same model training hyper-parameters and search space).

I also had trouble understanding Figure 2, which contains 8-10 overlapping red/blue lines. I also had trouble understanding the relationship between the "PC-DARTS" and "Best from Zela et al." lines (since Zela et al. evaluate PC-DARTS in their paper). It would be helpful to clarify the distinction between the two.


## Additional Notes

I had trouble understand this sentence from the end of Section 2.2: "Furthermore, for methods that adapt architecture parameters during search, it [using a first-order method] makes clear that we need not worry about rank disorder as long as we can optimize and generalize"
Why don't first-order methods need to worry about rank disorder? This needs to be better explained.

In Equation (5), what does the $\bigodot$ symbol denote? My best guess is that it refers to an elementwise product, but it would be helpful to state the meaning more clearly.

In Section 3.2: It's not clear to me what the significance of Theorem 1 is in practice. What is the significance of having a small expected value for $\Delta$ or of being close to an approximate stationary point? What does it mean, and why is it desirable? This needs to be better-explained.

---

> ### Author Response · Authors · 2020-11-17
> **Response to AnonReviewer1 (Part 2: notes and questions)**
>
> - _“[T]he improvements of GAEA algorithms over their non-GAEA equivalents seem quite small in most cases. (The authors attribute this to limited headroom in the search spaces, which seems plausible to me.) [..] The authors do not provide numbers for their own reproduction of the original PC-DARTS results; the writing implies that the hyper-parameters and search space are comparable. Providing numbers for the authors' reproduction of PC-DARTS would strengthen these results.”_
> Please see our response to concern 3 above regarding significance on the DARTS search space. We did not fully reproduce PC-DARTS results because we were able to recover the reported performance in initial tests and our settings are identical both in terms of the search space as well as the hyperparameter used for evaluation. Note that a full re-evaluation on this search space is very costly (close to $2,000 on AWS).
>
> - _“[T]he results presented in Figure 2 are difficult for me to interpret because the plots are visually noisy and contain many overlapping lines of different colors. [..] I also had trouble understanding the relationship between the "PC-DARTS" and "Best from Zela et al." lines (since Zela et al. evaluate PC-DARTS in their paper).”_
> In the revision we have slightly enlarged the plots and filled in the error bounds to have fewer individual lines. The “PC-DARTS” line is our reproduction of PC-DARTS on NAS-Bench-1Shot1, which performed better than the best result in Zela et al. (2020b) on search spaces 1 and 3 (this is why the dashed line is higher than the blue line in those two cases). We have clarified this point in the revised caption.
>
> - _“For other DARTS variants, the accuracy improvements are quite impressive (e.g., 94.10% accuracy for GAEA-DARTS-ERM), but the authors do not provide a non-GAEA baseline for this configuration. Furthermore, the accuracies that the original NASBench-201 paper provides for baselines such as ENAS and DARTS are extremely low, so I think that providing a careful reproduction of DARTS-ERM with properly tuned hyper-parameters would help make the results more convincing.”_
> Please see our response to concern 4.
>
> - _“I had trouble understanding Section 3 (which presents the paper's main theoretical results) in particular. The section is notation and terminology-heavy, and would be easier to follow if it spent more time explaining relevant terms.”_
> We have significantly expanded upon the discussion of stationarity in Section 3.2 of the revision.
>
> - _“I found Table 1 difficult to interpret because different rows of the table contain the results of architecture searches performed using different search spaces and/or model training hyper-parameters. For example, ProxylessNAS uses a very different search space than PC-DARTS and GAEA PC-DARTS. While the authors note this particular example in the text under the table, the table would be much easier to interpret if it was organized to show which subsets of the results were comparable to each other (i.e., used the same model training hyper-parameters and search space).”_
> In the revision we have divided Table 1 into methods evaluated on the DARTS search space (bottom) and on others (top). The only other option we see is to remove the top three methods entirely, but we believe they serve as a useful comparison. While evaluation routines for all methods on the DARTS search space are largely comparable, we specifically use the PC-DARTS routine, which differs slightly from DARTS (drop-path probability changed from 0.2 to 0.3). We have edited the caption to reflect this.
>
> - _“I had trouble understand this sentence from the end of Section 2.2: "Furthermore, for methods that adapt architecture parameters during search, it [using a first-order method] makes clear that we need not worry about rank disorder as long as we can optimize and generalize" Why don't first-order methods need to worry about rank disorder? This needs to be better explained.”_
> The “it” here is “the single-level formulation,” not “using a first-order method.” Our argument is that all (bilevel and single-level) weight-sharing methods (except those like RS-WS that do not update architecture parameters) do not need to worry about rank-disorder, since the main requirement is a well-generalizing solution to an optimization problem, which requires outputting a single point rather than a ranking. We have modified this discussion in the revision.
>
> - _“In Equation (5), what does the ⨀ symbol denote? My best guess is that it refers to an elementwise product, but it would be helpful to state the meaning more clearly.”_
> Yes, it stands for elementwise product. We have defined this in the revision.
>
> - _“In Section 3.2: It's not clear to me what the significance of Theorem 1 is in practice. What is the significance of having a small expected value for Δ or of being close to an approximate stationary point? What does it mean, and why is it desirable?”_
> Please see our response to concern 2.

---

> ### Author Response · Authors · 2020-11-17
> **Response to AnonReviewer1 (Part 1: main concerns)**
>
> Thank you for the detailed feedback. Below we respond to your questions/criticisms.
>
> Responses to main concerns:
> 1. _“Novelty is limited, since the authors are proposing a relatively limited change to an existing family of Efficient NAS algorithms. (This would, however, be a non-issue if the proposed changes were effective enough.)”_
> The novelty of the paper is not just in the empirical results and technical method but also in the optimization perspective on NAS with weight-sharing. As we argue below, we also believe our proposed changes are effective.
>
> 2. _“Unclear presentation of theoretical results.” “In particular: the paper claims to prove stationary-point convergence bounds for a variant of their algorithm but doesn't explain what stationary-point convergence means or why it's important.”_
> ε-stationary-point convergence is the standard measure of success in non-convex optimization; in the unconstrained Euclidean case it corresponds to the expected norm of the gradient being smaller than ε. This is significant because it implies we have reached a point that satisfies a necessary condition of global optimality (small gradient) and where most first-order algorithms do not make further progress, thus measuring the speed at which a first-order method effectively terminates. In the revision we have a substantially more detailed discussion of this in Section 3.2.
>
> 3. _“Quality improvements seem quite limited on 2 of the 3 benchmark tasks. (Part of the problem might -- as the authors suggest -- be due to limited headroom.)” “On two of the tasks, the performance of their modified algorithm seems quite close to that of a baseline without the modification. On the third task (NASBench-201), the improvements appear to be quite substantial in one case.”_
> We disagree that the improvement is small on the DARTS Search Space benchmark. While the changes on CIFAR-10 are small numerically due to the saturation on this dataset, the GAEA PC-DARTS average is one standard deviation better than the PC-DARTS average, and the best of 10 seeds is more than 3.5 standard deviations better than the PC-DARTS average; given the overall small remaining room for improvement and the fact that PC-DARTS was the state-of-the-art on this benchmark, we believe these results are important. Furthermore, we observe larger improvements on the ImageNet evaluation, especially in the setting of CIFAR-10 -> ImageNet transfer NAS, where GAEA PC-DARTS improves upon DARTS by 0.8% in terms of Top-1 error. We agree that on the NAS-Bench-1Shot1 search space the improvement due to GAEA is small, with existing algorithms already approaching optimality.
>
> 4. _“For the benchmark task with the largest improvement -- NASBench-201 -- the authors only provide their own baseline for 1 of the 3 NAS algorithms they evaluate, and for this variant (GDAS), the improvements over the baseline are quite small. So it's not clear to me how much of the improvements for the other two algorithms can be attributed to GAEA and how much can be attributed to NAS setups that are better tuned than the ones from the original NASBench-201 paper.”_
> We did not report our own reproductions of the other algorithms because we had no issues reproducing them, whereas we did for GDAS. Note that we did compare to our own evaluation of bilevel DARTS in the Appendix (Figure 4), showing that it exhibited the same poor performance. In revision we have added our own experiments for DARTS (ERM and bilevel) to Table 2; they show that the substantial improvements on both using GAEA are a result of our method and not different setups. Note that GAEA-based modifications of DARTS are not tuned on NAS-Bench-201: the only hyperparameter is the architecture learning rate, which we set to 0.1 for all DARTS/PC-DARTS experiments on all search spaces.

---

> ### Comment · AnonReviewer1 · 2020-11-17
> **Review Updates**
>
> I'd like to thank the authors for their detailed response to my original review. Based on the authors' feedback and their updates to the submission, I plan to update my score for the paper from 4 to 7.
>
> Here are the two biggest reasons for my score change:
>
> * The authors added accuracy numbers for their own reproductions of DARTS (ERM) and DARTS (bilevel) in Table 2. The addition of these baselines makes me much confidence in the effectiveness of the authors' proposed algorithmic modifications, since they allow me directly compare the results of the authors' GAEA DARTS results against the non-GAEA equivalents.
> * The authors improved their exposition of theoretical results, and added a few paragraphs in Section 3.2 to explain the significance of their $\epsilon$-stationary-point convergence results. (I'd also like to thank the authors for the explanation they provided in their OpenReview comments.)
>
> The authors also addressed many of my lower-level comments. For example:
>
> * Formally defining the $\odot$ symbol in Section 3.1 of the paper.
> * Cleaning up Figure 2 and adding additional information to its caption.
> * Clarifying in an OpenReview post that "GAEA-based modifications of DARTS are not tuned on NAS-Bench-201."

---

### Official Review · AnonReviewer3 · 2020-10-28
**Well written and SOTA methods.**

**Rating:** 8
**Confidence:** 4

**Review:**

This paper introduces the geometry-aware framework that can be adapted to any existing weight-sharing NAS methods optimized over gradient descent. The authors focus on the aspect of optimizing the architecture parameters to overcome the criticism of weight-sharing methods. The author's method relies on the mirror descent supporting their methods with a theoretical guarantee for the fast convergence. The author also supports their methods on various datasets such as CIFAR-10, ImageNet, NAS-Bench-201 (Dong & Yang), and NAS-Bench-1Shot (Zela et al.).

The paper supports their ideas not only supported by theoretical background, but outperforming results consistently with extensive empirical experiments such as CIFAR-10, ImageNet, NAS-Bench-1Shot1, and NAS-Bench-201. Furthermore, the GAEA can easily be applied to existing NAS methods which I believe making this work more valuable.

Strength
1. Their methods easily apply to existing NAS methods with gradient-based methods.
2. Theoretically supported methods with convergence guarantee.
3. Extensive empirical experiments on CIFAR-10, ImageNet, NAS-Bench-1Shot1, and NAS-Bench-201. Moreover, detailed experiment descriptions and fair setups that are critical in NAS comparison are provided.
4. Novel perspective of view (optimization perspective) to overcoming the weight-sharing methods' criticism of recent works.
5. Reproducible code included along with the paper.

Overall, I recommend clear acceptance. This paper will provide new insights/perspective to NAS algorithms which adopts weight-sharing methods.

Reference
1. Dong & Yang. NAS-Bench-201: Extending the Scope of Reproducible Neural Architecture Search (ICLR 2020)
2. Zela et al. NAS-Bench-1Shot1: Benchmarking and Dissecting One-Shot Neural Architecture Search (ICLR 2020)

---

> ### Author Response · Authors · 2020-11-17
> **Response to AnonReviewer3**
>
> Thank you for the positive feedback. We would be happy to answer any further questions stemming from the revision.

---

### Official Review · AnonReviewer2 · 2020-10-28
**This paper argues for using a single level objective on weight-sharing NAS, and proposes GAEA, which uses exponentiated gradient to update architecture parameters, to accelerate the convergence. The paper gives a proof to guarantee finite-time convergence. The experiment results show this method is efficient and can slightly improve the performance.**

**Rating:** 6
**Confidence:** 4

**Review:**

Pros:
1.	This paper gives a proof of finite-time convergence, which is the first paper working on this. Besides, the paper gives corresponding analysis of ENAS and DARTS. This is a new perspective of NAS methods.
2.	This work uses EG method to update architecture parameter, which takes the advantage of EG method and is reasonable to be applied on NAS problem.
3.	The experiment results show the efficiency and effectiveness of the proposed method.

Cons:
1.	Using EG to update architecture parameters can only accelerate convergence, which has nothing to do with improving the NAS performance. It is still confusing that why single-level optimization can resolve rank disorder and poor performance. It is not clear that the slight improvement on the performance is due to your algorithm or accident.
2.	Efficiency is claimed as an important point in this paper. However, only the results in Table 2 shows GAEA shorten the time cost. In Table 1, it cannot be detected that your method is more efficient. Is that because updating architecture parameters does not cost too much time in these experiment? If so, the contribution may be less.
3.	In Figure 2, it is not easy to detect the performance difference between your method and the baseline. You should also explain the meaning of lines with deeper colors.
4.	It is better to add discussion or conclusion at the end of your paper, which can help readers to better understand your work.

Overall Review:
This paper gives a theory about the convergence time of NAS methods, which provides new perspectives on NAS problem. The paper find that EG method is appropriate on updating architecture parameters and this method can improve the efficiency of NAS problem. There are also some questions mentioned above in this paper. With some modifications, this paper could be an excellent paper.

---

> ### Author Response · Authors · 2020-11-17
> **Response to AnonReviewer2**
>
> Thank you for the positive feedback. Below we respond to your questions/criticisms.
>
> 1.
>    - _“Using EG to update architecture parameters can only accelerate convergence, which has nothing to do with improving the NAS performance. [...] It is not clear that the slight improvement on the performance is due to your algorithm or accident.”_
> As we discuss in Sections 3.3 and 4.1, EG does not just affect convergence speed but is also known to encourage sparsity; in our case this means it is more likely to output sparse architecture parameters that do not require as much post-search discretization. We show in Figure 1 that NAS methods modified by GAEA do indeed yield sparser architecture parameters. Loss due to post-search discretization is a well-known issue of DARTS-like NAS methods, and we give quantitative evidence in Section 4.1 that the drop in validation accuracy due to discretization is indeed smaller for GAEA.
>    - _“It is still confusing that why single-level optimization can resolve rank disorder and poor performance.”_
> Our argument in Section 2 is not that single-level optimization resolves rank disorder, but rather that we do not need to worry about rank disorder so long as we can use optimization and regularization techniques to output a single good architecture. The usefulness of single-level optimization is (1) making this point more clear by expressing weight-sharing NAS as a regular deep learning optimization problem needing only a point solution and (2) serving as a useful object for theoretical study. We have modified our discussion of this in the revision.
>
> 2. _“Efficiency is claimed as an important point in this paper. However, only the results in Table 2 shows GAEA shorten the time cost. In Table 1, it cannot be detected that your method is more efficient. Is that because updating architecture parameters does not cost too much time in these experiment? If so, the contribution may be less.”_
> On NAS-Bench-201 (Table 2), we found that GAEA-modified methods found good architectures much faster than the original methods and could be stopped early, leading to speedups. On the DARTS Search Space (Table 1), we found that both PC-DARTS and GAEA PC-DARTS yield good architectures when stopped early (GAEA is somewhat better earlier on, but not significantly so), so for fairness and consistent comparison with previous papers we used the same number of search epochs in both cases. Finally, note that most supernet training procedures are effectively early-stopped (as evidenced by fewer training epochs during architecture search than architecture evaluation) since the network has to be retrained anyway, so the faster EG convergence likely manifests itself in better final accuracy since the supernet is closer to convergence at the end of search: we give evidence of this in Section 4.1, where the GAEA PC-DARTS supernet has a 2.9% better final validation accuracy than the PC-DARTS supernet, and in Figure 4 of the appendix, which shows that GAEA DARTS converges on NAS-Bench-201 while DARTS does not.
>
> 3. _“In Figure 2, it is not easy to detect the performance difference between your method and the baseline. You should also explain the meaning of lines with deeper colors.”_
> In the revision we have slightly enlarged the plots, filled in the error bounds to have fewer individual lines, and explained the meaning of the different lines in the caption.
>
> 4. _“It is better to add discussion or conclusion at the end of your paper, which can help readers to better understand your work.”_
> Thank you for the suggestion; we have added a conclusion in the revision.

---

> > ### Comment · AnonReviewer2 · 2020-11-24
> > **Response**
> >
> > Thank the authors for the detailed response to my original review. The explanation has clearly answer some of my concerns. In my opinion, the paper mainly states three things before the experiment: single-level NAS, mirror descent and the theorem of convergence. The paper discusses why single-level NAS is good. Although the algorithm is similar with XNAS, the paper gives another perspective to interpret the algorithm by mirror descent way. The paper also provides a theorem of convergence of NAS problem.
> > However, I think the problem here is the is no strong connections stated among the three things. Lots of NAS algorithms are single level. Since \phi in assumption 1 is general, the theorem 1 can be applied on other methods as well (if I understand correctly), such as single-level DARTS. These two parts seems separated with GAEA, and there is no story provided to smoothly contact these parts together, which makes your paper not so smooth, and readers cannot easily understand the main contribution of your paper.

---

> > > ### Author Response · Authors · 2020-11-24
> > > **Response to AnonReviewer2**
> > >
> > > Thank you for your follow-up message.
> > >
> > > We would like to clarify the connection between single-level NAS, mirror descent, and our convergence result. First, note that we argue in favor of studying the single-level algorithm in theory, not that it should be better in practice. It is by studying the single-level optimization problem that we are able to extend recent results in non-convex mirror descent and obtain a convergence guarantee.
> > >
> > > As you correctly point out, this convergence guarantee can be used for a variety of algorithms. However, it is also prescriptive, in that it suggests using a method that has better guarantees over the geometry of interest; for the simplex case that arises in NAS the suggested method is GAEA. Thus our single-level analysis is directly connected to our proposed method.
> > >
> > > Of course, as with any such analysis this approach has its limitations in practice; we discuss this in point 2 of our response to your original review. Still, this type of algorithmic comparison is not possible using results in existing work such as XNAS, and our geometric understanding of the problem enables the application of GAEA to numerous NAS methods, not just one.

---

### Author Response · Authors · 2020-11-16
**Revision summary**

We thank all three reviewers for thoughtful reviews. We have uploaded a revision incorporating the feedback, including the following changes:

- Improved readability of Figure 2 (following comments by reviewers 1 and 2).
- Expanded discussion in Sections 2 and 3 (following comments by reviewers 1 and 2).
- Cleaning up the presentation of Table 1 (following comments by reviewer 1).
- Additional baseline results in Table 2 (following comments by reviewer 1).
- A conclusion section (following comments by reviewer 2).
- Minor fixes.

We are happy to answer any further questions and would appreciate additional feedback.

---

### Decision · Program_Chairs · 2021-01-07
**Final Decision**

**Decision:**

Accept (Spotlight)

**Comment:**

This is a solid paper providing the first theoretical convergence result for NAS and showing promising empirical results.
The authors' proposed GAEA method can be combined with different types of weight-sharing algorithms (DARTS, PC-DARTS, etc) and is likely to reduce the impact of the architecture discretization step due to finding sparser solutions.
I clearly recommend acceptance and would expect this to make a nice spotlight.